# Optimal Pricing in Repeated Posted-Price Auctions with Different Patience of the Seller and the Buyer

**Arsenii Vanunts**
Yandex
Moscow, Russia
avanunts@yandex.ru

**Alexey Drutsa**
Yandex; MSU
Moscow, Russia
adrutsa@yandex.ru

## Abstract

We study revenue optimization pricing algorithms for repeated posted-price auctions where a seller interacts with a single strategic buyer that holds a fixed private valuation. When the participants non-equally discount their cumulative utilities, we show that the optimal constant pricing (which offers the Myerson price) is no longer optimal. In the case of more patient seller, we propose a novel multidimensional optimization functional — a generalization of the one used to determine Myerson's price. This functional allows to find the optimal algorithm and to boost revenue of the optimal static pricing by an efficient low-dimensional approximation. Numerical experiments are provided to support our results.

## 1 Introduction

Auctions have been studied for decades [82] and remain the main instrument for extracting revenue in Internet advertising for many years [36]. Revenue optimization problem in static (i.e., one-period) auctions is well studied and has proved its great worth to the Internet industry [64, 2], while the same problem in dynamic auctions is still understudied [57], although the major part of web advertisement sales has repeated nature [7, 32]. Consider the following example: an RTB platform (*a seller*) tracks a user and repeatedly sells impressions on the user's screen to advertisers (*buyers*) until the user is out of the RTB's sight. This example is naturally modeled by a sequence of repeated auctions, in which buyers have *fixed valuation* for a good all the way through.

For more than eleven years generalized second-price (GSP) auctions remain the leading instrument for selling ads [79] and, as argued by [7, 8, 61, 30, 32, 33], a significant part of auctions in AdExchanges involve only a single buyer. Single-buyer GSP auctions are known in the literature as *posted-price auctions* [51]. Repeated setting of them is referred to as *repeated posted-price auctions* in studies on worst-case regret minimization [8, 32] and as *a fishmonger's problem* in studies on expected revenue maximization [29]. The setting of the fishmonger's problem relies on the assumption that the seller knows the distribution of the *buyer valuation* of a good. This assumption is realistic for advertising auctions, since most Internet companies possess rich historical bidding data [58, 67].

We study the fishmonger's problem in which the seller repeatedly sells goods through a posted-price mechanism to the *same* buyer that holds a *fixed private* valuation for a good. The buyer seeks to maximize his cumulative surplus, which is a discounted sum of his instant utilities over all rounds. The seller knows the valuation distribution and the buyer's discount sequence; so, she applies a pricing algorithm that sets prices in each round in order to learn the valuation and extract more revenue. The algorithm is announced to the buyer in advance [61], thus, the buyer picks an optimal strategy w.r.t. the announced algorithm, the valuation, and his discount sequence. The seller optimizes her expected cumulative revenue — a discounted sum of her instant expected utilities over all rounds — w.r.t. her discount sequence, the valuation distribution, and the buyer's discount sequence.

When both the seller and the buyer equally discount their utilities, an optimal pricing is known from a "folklore wisdom" [29]: it is the constant pricing algorithm that proposes the Myerson optimal price [64] each round. Thus, the seller cannot advantageously apply any dynamic learning of prices (based on previous decisions of the buyer) to improve her revenue with respect to a much simpler approach that offers the optimal static constant price over all rounds. However, in many real applications, the equal discount assumption may not hold due to an imbalance between the sides in the patience to wait for utility [7, 8, 61] or their ability to estimate the probability that the game do not terminates in a round [75, 61]. The case of our setting where *the time discounts of the seller and the buyer are different* was never studied before[1]. In this work, we attempt to fill this gap.

In the case of less patient seller (i.e., the seller discount is less than the buyer one), we show that the "folklore wisdom" technique can be easily adapted to prove that "Big deal" algorithm[2] is optimal. The expected revenue of this pricing is shown to be strictly greater than the one of the optimal constant algorithm, if the seller is strictly less patient. In the inverse case (the buyer is less patient), we show that the problem is much more challenging and cannot be resolved by the "folklore wisdom" or Myerson techniques. The problem in the initial form has structure similar to a saddle-point problem: the revenue depends on an algorithm via argmax over buyer strategies and the derivatives of such dependence have exponential number of jump discontinuities (Sec. 4). Hence, the initial revenue optimization problem can be numerically solved only via a brute-force search.

In our work, first, for the game with a finite horizon $T$, we reduce the problem to *the optimization of a novel multivariate functional* (Th. 3) that constitutes a generalization of the one used to determine Myerson's price. This functional has a simple bilinear-like structure and is continuously differentiable as many times as the CDF of the valuation distribution. This allows to find the optimal pricing algorithm by means of a variety of efficient gradient-based methods. Second, for any game, we make a low-dimensional approximation of the optimal revenue problem by an optimal $\tau$-*step algorithm*[3], which can be found using our reduction approach as well (Sec. 5). In this way, *our multivariate functional constitutes a powerful and simple technique that allows the seller to significantly increase her revenue* (w.r.t. the optimal static pricing) even in the games with large $T$. So, we provide the rule of thumb: choose $\tau$ to fit your computational capabilities (e.g., $\tau = 2,3$), find the optimal $\tau$-step pricing by the functional, and apply the prices learned in this way to get a boost in revenue.

Finally, we support our findings by an extensive numerical experimentation for a variety of discount rates. We demonstrate that optimal algorithms are non-trivial, may be non-consistent [30, 32], have prices noticeably dependent on the discounts, and generate revenue larger than the constant algorithm with Myerson's price (Sec. 5). Overall, *our main contribution* is our reduction approach that allows both to find the optimal algorithm (even with possible structural constraints) and to boost revenue by the efficient low-dimensional approximation in the case of less patient buyer.

## 2    Problem statement and preliminaries

**Setup.** A single seller and a single buyer interact repeatedly over a sequence of $T$ rounds, where *the horizon $T$ is either finite, or infinite*. The seller possess a fresh copy of a good each round and the buyer values each copy of this good by *a fixed private valuation $v \in \mathbb{R}_+$*. At each round $t \in [T] := \{1, ..., T\}$ the seller sets a price $p_t$ for a new copy of the good and the buyer answers with a decision $a_t \in \{0, 1\}$: an accept 1 or a reject 0. Sequences of the buyer's answers are denoted by bold Latin letters, e.g., $\mathbf{a} = \{a_t\}_{t=1}^T$, and are referred to as *buyer strategies*. The price $p_t$ depends on the previous answers of the buyer $a_1, .., a_{t-1}$, i.e. the seller uses a deterministic *pricing algorithm $\mathcal{A}$* to set prices [30].

Given an algorithm $\mathcal{A}$ and a strategy $\mathbf{a}$, the price sequence $\{p_t\}_{t=1}^T$ is uniquely determined. The *instant utilities* of the buyer and the seller are $a_t(v - p_t)$ and $a_t p_t$, respectively, in round $t$. The instant utilities contribute to the buyer's (the seller's) total utility w.r.t. a discount $\gamma_t^{\mathrm{B}}$ ($\gamma_t^{\mathrm{S}}$, resp.). Total utilities of the buyer and the seller are referred to as *the buyer surplus* and *the seller revenue*: $\mathrm{Sur}_{\gamma^{\mathrm{B}}}(\mathcal{A}, \mathbf{a}, v) := \sum_{t=1}^T \gamma_t^{\mathrm{B}} a_t(v - p_t)$ and $\mathrm{Rev}_{\gamma^{\mathrm{S}}}(\mathcal{A}, \mathbf{a}) := \sum_{t=1}^T \gamma_t^{\mathrm{S}} a_t p_t$, resp. Both the buyer and the seller are rational and risk-neutral agents [52]. *Discount sequences $\gamma^{\mathrm{B}} = \{\gamma_t^{\mathrm{B}}\}_{t=1}^T$ and $\gamma^{\mathrm{S}} = \{\gamma_t^{\mathrm{S}}\}_{t=1}^T$*

are positive, $\gamma_t^{\text{B}}, \gamma_t^{\text{S}} > 0 \ \forall t$, and have finite sums: $\Gamma^{\text{B}} := \sum_{t=1}^{T} \gamma_t^{\text{B}}, \Gamma^{\text{S}} := \sum_{t=1}^{T} \gamma_t^{\text{S}} < \infty$. For simplicity of presentation, from here on in our paper we assume that the discounts decrease geometrically: $\gamma_t^{\text{B}} = \gamma_{\text{B}}^{t-1}$ and $\gamma_t^{\text{S}} = \gamma_{\text{S}}^{t-1}$ for some $\gamma_{\text{B}}, \gamma_{\text{S}} > 0$. However, our results hold for a larger variety of discounts (see Remarks 1 and 2).

Our setting is based on two standard assumptions: (1) the buyer knows the pricing algorithm $\mathcal{A}$ in advance (i.e., the seller commits to it at the beginning of the first round); and (2) the seller knows the distribution $D$ from which the private buyer valuation $v$ (unknown to her) is drawn. Assumption (1) matches the practice in Internet advertising [61] since RTB platforms run hundreds of millions auctions a day [7, 30] (see App. F for mode details). Assumption (2) is realistic since most Internet companies have access to rich historical data [67]. We also assume that the seller knows the exact buyer discount sequence.[4] The CDF and the density of $D$ is denoted by $F_D$ and $f_D$, resp. The random valuation is denoted by $V, V \sim D$.

Our rational buyer with the private valuation $v$ that knows the algorithm $\mathcal{A}$ in advance is referred to as *a strategic buyer* [7] and exploits *an optimal strategy* $\mathbf{a}^{\text{Opt}}(\mathcal{A}, v, \boldsymbol{\gamma}^{\text{B}}) := \text{argmax}_{\mathbf{a} \in \mathfrak{S}_T} \text{Sur}_{\boldsymbol{\gamma}^{\text{B}}}(\mathcal{A}, \mathbf{a}, v)$, where $\mathfrak{S}_T := \{0, 1\}^T$ is the set of all possible strategies. This leads us to the definition of the *strategic revenue* of the pricing $\mathcal{A}$, which faces the strategic buyer with a valuation $v$:

$$\text{SRev}_{\boldsymbol{\gamma}^{\text{S}}, \boldsymbol{\gamma}^{\text{B}}}(\mathcal{A}, v) := \text{Rev}_{\boldsymbol{\gamma}^{\text{S}}}(\mathcal{A}, \mathbf{a}^{\text{Opt}}(\mathcal{A}, v, \boldsymbol{\gamma}^{\text{B}})). \tag{1}$$

We consider the problem of pricing optimization from the seller's point of view. This problem is stated as follows: find such algorithm $\mathcal{A}^*$ that its *expected strategic revenue (ESR)* $\mathbb{E}_{V \sim D}[\text{SRev}_{\boldsymbol{\gamma}^{\text{S}}, \boldsymbol{\gamma}^{\text{B}}}(\mathcal{A}^*, V)]$ is not less than the ESR of any other algorithm (i.e. the ESR of the algorithm $\mathcal{A}^*$ is the maximum).

**Notations and auxiliary definitions.** Following [51, 61, 30, 32], we associate a deterministic pricing algorithm with a perfect $(T{-}1)$-depth binary tree with labeled nodes. Let $\mathfrak{N}_T$ be the set of nodes of the tree and $\mathcal{A}(\mathfrak{n})$ be a label of a node $\mathfrak{n} \in \mathfrak{N}_T$. In the first round, *the current node* is the root $\mathfrak{e} \in \mathfrak{N}_T$. Let $\mathfrak{n}$ be the current node in a round $t$ (the depth $|\mathfrak{n}|$ of $\mathfrak{n}$ is $t - 1$); then the algorithm offers the price $\mathcal{A}(\mathfrak{n})$. If this price is rejected, the current node moves to the $\mathfrak{n}$'s left child denoted by $\mathfrak{n}0$, otherwise the current node moves to the $\mathfrak{n}$'s right child denoted by $\mathfrak{n}1$. We denote nodes by finite strings over the alphabet $\{0, 1\}$: the root is the empty string $\mathfrak{e}$, its left child is $0$, the right one is $1$, the right child of $0$ is $01$, etc. (e.g. $0^k$ is the string of $k$ zeros). Thus, $\mathfrak{N}_T := \{\mathfrak{n} \in \{0, 1\}^* \mid |\mathfrak{n}| < T\}$, where $|\mathfrak{n}|$ is the length of the string $\mathfrak{n}$, and the set of algorithms $\mathfrak{A}_T$ is the set of maps from $\mathfrak{N}_T$ to $\mathbb{R}_+$: $\mathfrak{A}_T = \mathbb{R}_+^{\mathfrak{N}_T}$.

**Research questions.** One of standard interpretations[5] of a discount factor $\gamma_{\text{B}}^{t-1}$ (or $\gamma_{\text{S}}^{t-1}$) is the participant's estimate of the probability that the repeated auctions will last at least $t$ rounds [29, 32]. While the *constant Myerson algorithm* (see Sec. 3) is a well-known folklore solution for the case of equal discounts [29], the case of different discounts, in our setup, was never considered earlier, although it is more realistic. In Internet advertising, the seller and the buyer are usually companies of different sizes, with different opportunities and capabilities (e.g., an RTB platform vs. a web site with an advertisement, see App. F for an example as well). In this way, they may have different data (or access to them) that are used to make an estimation of the game-continuation probability (i.e., the discount factor). For instance, most likely that the RTB platform has more data and may know which data are not available to the advertiser. As a result, the auction participants have different discounts.

In the case when the buyer overestimates the discount factor $\gamma_{\text{B}} > \gamma_{\text{S}}$, we show that the seller can obtain $(1 - \gamma_{\text{S}})/(1 - \gamma_{\text{B}})$ times larger expected revenue than the one of the constant Myerson algorithm: she should apply the "Big deal" pricing algorithm (Sec.3). The inverse case appears to be non-trivial, and, in our study, we primarily address the following research questions in the case of $\gamma_{\text{S}} > \gamma_{\text{B}}$ (Sec. 4 and 5): (1) *What is the optimal algorithm and its expected strategic revenue?* (2) *How much more is the maximal ESR than the constant Myerson's one?* (3) *Can the seller extract expected revenue more than in the static Myerson pricing having limits on computational resources?*

**Related work.** There are two series of works that are most relevant to ours. The first one studied repeated posted-price auctions in the worst-case scenario [51, 7, 8, 61, 30, 31, 32], where our setting of the strategic buyer with a fixed private valuation is considered. Amin et al. [7] proposed to seek for algorithms that have the lowest possible asymptotic upper bound on the strategic regret for the *worst case* valuation of the buyer. Recently, Drutsa [30, 32, 33] has found pricings with optimal regret

bound. In contrast to these studies, first, we search for a pricing algorithm that maximizes the strategic revenue *expected* over buyer valuations, which matches the practice of ad exchanges and optimization goals in classical auction theory [52]. Second, our revenue optimization problem is solved *exactly* (not via optimization of lower/upper bounds). Third, our study considers a more general setup in which not only the buyer's surplus is discounted over rounds, but also the seller's revenue does. The second series studied repeated posted-price auctions with an incomplete information and in the absence of the ability to commit [43, 75, 29, 47]. The authors of [43, 75] showed that, in the case of non-commitment, the seller is forced to sell a good for a minimal possible price until last few rounds. Devanur et al. [29] showed that the seller can obtain non-trivial revenue, if she is able to partially commit, e.g. to commit not to raise prices. Enhancing the competition was shown to allow the seller to extract non-trivial revenue as well [47]. However, all these works treated the "commitment" revenue as a unachievable benchmark. Hence, in our case of repeated auctions that is motivated by Internet advertisement sales, where the seller is able to commit[6], it is unreasonable to consider the non-commitment case. Our setting (but in the case of equal discounts) can be considered as a more general dynamic mechanism design problem studied, e.g. in [49, 71, 70, 13]. To the best of our knowledge this line of work never considered scenarios with different discounts. It would be a great future study to generalize our results for different discounts to more general dynamic mechanisms.

## 3 Less patient seller: the case of $\gamma_{\mathsf{S}} \leq \gamma_{\mathsf{B}}$

Our study begins with the analysis of the case $\gamma_{\mathsf{S}} \leq \gamma_{\mathsf{B}}$ in two steps. First, the subcase of equal discounts $\gamma_{\mathsf{S}} = \gamma_{\mathsf{B}}$ can be resolved by means of the classical auction theory [64]. Second, we reduce the whole case $\gamma_{\mathsf{S}} \leq \gamma_{\mathsf{B}}$ to the subcase $\gamma_{\mathsf{S}} = \gamma_{\mathsf{B}}$ by showing that, for $\gamma_{\mathsf{S}} \leq \gamma_{\mathsf{B}}$, the seller can obtain the same strategic revenue as if her discount was $\gamma_{\mathsf{B}}$ instead. Two simple optimal algorithms are provided.

**Equal discounts: a constant algorithm.** Let $\gamma_{\mathsf{S}} = \gamma_{\mathsf{B}} = \gamma$, then one can apply the almost folklore technique of reducing this subcase to a single-round feasible mechanism [29]. Key steps of this technique are provided in App. A.1.1 for completeness of our study on different discounts. The expected revenue of the obtained feasible mechanism is known [64] to be no greater than $p_D^*(1 - F_D(p_D^*))$, where $F_D$ is the CDF of our valuation variable $V \sim D$ and $p_D^*$ is the *Myerson price*, i.e., the price that maximizes the functional $H_D(p) := p\mathbb{P}[V \geq p] = p(1 - F_D(p))$[7]. Thus, the following upper bound holds: $\mathbb{E}\left[\mathrm{SRev}_{\gamma,\gamma}(\mathcal{A}, V)\right] \leq \Gamma p_D^*(1 - F(p_D^*)) \; \forall \mathcal{A} \in \mathfrak{A}_T$. This bound is achieved, in particular, by the algorithm $\mathcal{A}_1^*$ which constantly offers the price $p_D^*$, i.e., $\forall \mathfrak{n} \; \mathcal{A}_1^*(\mathfrak{n}) = p_D^*$, and is referred to as the *optimal constant algorithm*. Overall, the following theorem holds:

**Theorem 1** ([29]). *Let the discount rates be equal: $\gamma_{\mathsf{S}} = \gamma_{\mathsf{B}} = \gamma$. Then the optimal constant algorithm $\mathcal{A}_1^*$ is optimal among all pricing algorithms $\mathfrak{A}_T$ and the optimal revenue is $\Gamma p_D^*(1 - F(p_D^*))$.*

**"Big deal" for less patient seller.** Let us consider the whole case of less patient seller: $\gamma_{\mathsf{S}} \leq \gamma_{\mathsf{B}}$. It is easy to see that $\mathrm{Rev}_{\gamma_1}(\mathcal{A}, \mathbf{a}) \leq \mathrm{Rev}_{\gamma_2}(\mathcal{A}, \mathbf{a})$ for any $\mathcal{A}$ and $\mathbf{a}$, when $\gamma_1 \leq \gamma_2$. Hence, for any $\mathcal{A}$, $v$, and $\gamma_1 \leq \gamma_2$, the inequality $\mathrm{SRev}_{\gamma_1, \gamma^{\mathsf{B}}}(\mathcal{A}, v) \leq \mathrm{SRev}_{\gamma_2, \gamma^{\mathsf{B}}}(\mathcal{A}, v)$ holds as well, since the optimal strategy $\mathbf{a}^{\mathrm{Opt}}$ does not depend on the seller's discount $\gamma^{\mathsf{S}}$. So, taking $\gamma_1 = \gamma_{\mathsf{S}}$ and $\gamma_2 = \gamma_{\mathsf{B}}$, one gets:

$$\max_{\mathcal{A} \in \mathfrak{A}_T} \mathbb{E}\left[\mathrm{SRev}_{\gamma^{\mathsf{S}}, \gamma^{\mathsf{B}}}(\mathcal{A}, V)\right] \leq \max_{\mathcal{A} \in \mathfrak{A}_T} \mathbb{E}\left[\mathrm{SRev}_{\gamma^{\mathsf{B}}, \gamma^{\mathsf{B}}}(\mathcal{A}, V)\right] = \Gamma^{\mathsf{B}} p_D^*(1 - F_D(p_D^*)). \tag{2}$$

The latter identity in Eq. (2) is from Th. 1. The bound in Eq. (2) is achievable as well. Namely, let us consider the following algorithm $\mathcal{A}_{\mathsf{bd}}^*$ (referred to as the "*big deal*") given $\gamma_{\mathsf{B}}$ and $V \sim D$: the first price is $\mathcal{A}_{\mathsf{bd}}(\mathfrak{e}) = \Gamma^{\mathsf{B}} p_D^*$; if the buyer accepts it, prices in further rounds will be $\mathcal{A}_{\mathsf{bd}}(1 \circ \mathfrak{n}) = 0 \; \forall \mathfrak{n}$; otherwise $\mathcal{A}_{\mathsf{bd}}(0 \circ \mathfrak{n}) = p_D^* \; \forall \mathfrak{n}$. An attentive reader may note that the strategic buyer accepts the first price $\mathcal{A}_{\mathsf{bd}}^*(\mathfrak{e}) \Leftrightarrow v > p_D^*$. Hence, similarly to the algorithm $\mathcal{A}_1^*$, it is easy to show that the ESR of $\mathcal{A}_{\mathsf{bd}}^*$ is $\Gamma^{\mathsf{B}} p_D^*(1 - F(p_D^*))$. The key idea behind the algorithm $\mathcal{A}_{\mathsf{bd}}^*$ is quite simple. Roughly speaking, the seller "accumulates" all her revenue at the first round by proposing the buyer a "big deal" that incentivises him to pay a large price at the first round and get all goods in the subsequent rounds for free, or, otherwise, get nothing[8]. Overall, the following theorem holds:

**Theorem 2.** *Let the discount rates be s.t. $\gamma_{\mathsf{S}} \leq \gamma_{\mathsf{B}}$. Then the "big-deal" algorithm $\mathcal{A}_{\mathsf{bd}}^*$ is optimal among all pricing algorithms $\mathfrak{A}_T$ and the optimal revenue is $\Gamma^{\mathsf{B}} p^*(1 - F_D(p^*))$.*

Th. 2 implies that, first, *the optimal constant algorithm $\mathcal{A}_1^*$ is not the unique optimal one* in the subcase of equal discounts $\gamma_S = \gamma_B$. Second, in the other subcase of $\gamma_S < \gamma_B$, the constant algorithm $\mathcal{A}_1^*$ is no longer optimal: the relative ESR of the optimal algorithm $\mathcal{A}_{bd}^*$ w.r.t. the optimal constant one $\mathcal{A}_1^*$ is $\Gamma^B / \Gamma^S$, which is $> 1$, when $\gamma_S < \gamma_B$[9]; i.e. *the optimal revenue is larger than the one obtained by offering the Myerson price constantly.* This result is quite inspiring for the seller, since the dominance of the buyer's discount $\gamma_B$ over the seller's one $\gamma_S$ suggests a hypothesis that the seller should earn lower than with $\gamma_B$ (e.g., see the revenue of $\mathcal{A}_1^*$). But the ability of the seller to apply the trick of "accumulation" of all her revenue at the first round allows her to get the payments for all goods discounted by the buyer's $\gamma_B$ at the first round and to boost thus her revenue over the constant pricing.

**Remark 1.** All results of the section hold even for non-geometric discounts s.t. $\boldsymbol{\gamma}^S \leq \boldsymbol{\gamma}^B$ (see App.A.1).

## 4 Less patient buyer ($\gamma_S \geq \gamma_B$): reduction to an optimization functional

This section provides the central fundamental results of our study. They are obtained for finite games, but, further, we show how to use them to get approximately optimal algorithms even for infinite games. In contrast to the case $\gamma_S \leq \gamma_B$, finding an optimal pricing for $\gamma_S \geq \gamma_B$ is much more difficult problem since the technique used in Sec. 3 to upper bound the expected strategic revenue is no longer applicable (because it relies on the condition $\gamma_S \leq \gamma_B$) and a generalization of the functional $H_D(\cdot)$ to a multivariate analogue is required. Note that the optimization problem of the ESR has structure similar to a saddle-point problem: the ESR depends on $\mathcal{A}$ via $\mathbf{a}^{Opt}$ which is an argmax over the set of strategies $\mathfrak{S}_T$. Moreover, the derivative of such dependences are piecewise continuous with jump discontinuities on the boundaries of pieces (there are $2^{2^T-3}$ pieces with derivatives of different forms). Hence, the problem in the initial form can be numerically solved only via brute-force search.

In order to make numerical solution of the problem more feasible, we will reduce it into the form of a multidimensional maximization of a simple bilinear-like function (namely, $L(\mathbf{v})$ in Eq.(4)) that is continuously differentiable as many times as the CDF $F_D$ and its derivatives have simple form and can be easily computed. The key steps are: (1) find a class of algorithms whose prices (2) can be linearly parametrized by points in the support of $D$ s.t. (3) the strategic revenue is constant between these points. For the sake of presentation, we consider regular discounts.

**Definition 1.** A discount sequence $\boldsymbol{\gamma}$ is *regular*, if $\boldsymbol{\gamma} \cdot \mathbf{a}^1 \neq \boldsymbol{\gamma} \cdot \mathbf{a}^2$ for any strategies $\mathbf{a}^1, \mathbf{a}^2 \in \mathfrak{S}_T$, i.e., any buyer strategy $\mathbf{a} \in \mathfrak{S}_T$ results in a unique discounted quantity of purchased goods ($\mathbf{a} \cdot \mathbf{b} := \sum_t a_t b_t$).

**Definition 2.** Let $\boldsymbol{\gamma}$ be a discount, then an algorithm $\mathcal{A} \in \mathfrak{A}_T$ is said to be *completely active (CA) for* $\boldsymbol{\gamma}$, if for any strategy $\mathbf{a} \in \mathfrak{S}_T$ there exists a valuation $v \in \mathbb{R}_+$ s.t. $S_{\mathbf{a}}(v) = S(v)$, where $S_{\mathbf{a}}(v) := \mathrm{Sur}_{\boldsymbol{\gamma}}(\mathcal{A}, \mathbf{a}, v)$ and $S(v) := S_{\mathbf{a}^{Opt}(\mathcal{A}, v, \boldsymbol{\gamma})}(v)$, i.e., the surplus function $S_{\mathbf{a}}$ (as a line) is tangent to the optimal surplus function $S$. We denote the set of all CA algorithms for $\boldsymbol{\gamma}$ by $\tilde{\mathfrak{A}}_T(\boldsymbol{\gamma})$.

A CA algorithm is such that any node in its labeled tree can be reached by the strategic buyer for at least one valuation $v$, i.e., be active. Surprisingly, *any algorithm can be transformed to a completely active one for* $\boldsymbol{\gamma}^B$ *with no loss in the expected strategic revenue.* Indeed, let $\mathcal{A}$ be a non-CA algorithm for $\boldsymbol{\gamma}^B$, then there exists *an inactive strategy* $\mathbf{a} \in \mathfrak{S}_T$ (i.e. $\forall v \geq 0$ $S_{\mathbf{a}}(v) < S(v)$). We tune $\mathcal{A}$ in such a way that $S_{\mathbf{a}}$ becomes tangent to $S$ without affecting the other surplus functions $S_{\mathbf{b}}$ for $\mathbf{b} \neq \mathbf{a}$ (it is visualized in Fig. A.1 in App. A.2.1). Namely, let $\tau$ be the index of the last $1$ in $\mathbf{a}$ and $\mathfrak{n} := \mathbf{a}_{1:\tau-1}$ be the $(\tau-1)$-round substrategy of $\mathbf{a}$. We decrease $p := \mathcal{A}(\mathfrak{n})$ until $S_{\mathbf{a}}$ becomes tangent to $S$. This operation will move also all $S_{\mathbf{b}}$ s.t. $\mathbf{b}_{1:\tau} = \mathbf{a}_{1:\tau}$ to the left. In order to make them unaffected, we simultaneously increase $p_s := \mathcal{A}(\mathfrak{n} \circ 10^s)$ for $0 \leq s \leq T - \tau - 1$ in such a way that $p + \gamma_B^{s+1} p_s = const$. Hence, $\mathbf{a}^{Opt}(\mathcal{A}, v, \boldsymbol{\gamma}^B)$ is unaffected for all $v$ except the point of tangency. Since $\gamma_S > \gamma_B$, the revenues $\mathrm{Rev}_{\boldsymbol{\gamma}^S}(\mathcal{A}, v, \mathbf{b})$ only increase after our tuning, when $\mathbf{b}_{1:\tau} = \mathbf{a}_{1:\tau}$, otherwise they are not changed for $\mathbf{b} \neq \mathbf{a}$ what infers that $\mathrm{SRev}_{\boldsymbol{\gamma}^S, \boldsymbol{\gamma}^B}(\mathcal{A}, \cdot)$ increases in all points except one. Tuning of the algorithm by "activating" all inactive strategies one by one in descending order of $\tau$ (this ensures that decreasing of $p$ will not result in negative prices) gives us a CA (for $\boldsymbol{\gamma}^B$) algorithm without loss in the ESR. Formally, the following proposition holds (the proof is in App. A.2.1).

**Proposition 1.** *Let $T \in \mathbb{N}$ and $\gamma_S, \gamma_B$ be discount rates s.t. $\gamma_S \geq \gamma_B$ and the sequence $\boldsymbol{\gamma}^B = \{\gamma_B^{t-1}\}_{t=1}^T$ is regular. Then, for any pricing algorithm $\mathcal{A} \in \mathfrak{A}_T$, there exists a CA algorithm $\tilde{\mathcal{A}} \in \tilde{\mathfrak{A}}_T(\boldsymbol{\gamma}^B)$ s.t.*

$$\mathbb{E}\big[\mathrm{SRev}_{\boldsymbol{\gamma}^S, \boldsymbol{\gamma}^B}(\mathcal{A}, V)\big] \leq \mathbb{E}\big[\mathrm{SRev}_{\boldsymbol{\gamma}^S, \boldsymbol{\gamma}^B}(\tilde{\mathcal{A}}, V)\big]. \tag{3}$$

The fundamental property of a CA algorithm: *it bijectively corresponds to the break (discontinuity) points of the derivative of its surplus function $S(\cdot)$, which is piecewise linear*[10]. Namely, the class $\tilde{\mathfrak{A}}_T$ can be linearly mapped onto $\Delta^k := \{\mathbf{v} = \{v_j\}_{j=1}^k \in \mathbb{R}^k | 0 \leq v_1 \leq \ldots \leq v_k\}$, where $k := k(T) := 2^T - 1$. The key intuition is as follows. Number the buyer strategies $\mathfrak{S}_T = \{\mathbf{a}^0, \ldots, \mathbf{a}^k\}$ in ascending order of the slope $\boldsymbol{\gamma}^{\mathrm{B}} \cdot \mathbf{a}^i$ of the corresponding $\boldsymbol{\gamma}^{\mathrm{B}}$-discounted surplus function $S_{\mathbf{a}^i}$ (the $\boldsymbol{\gamma}^{\mathrm{B}}$-*dependent natural order*). Let $(v_i, s_i)$ be the coordinates of the intersection of the straight lines $S_{\mathbf{a}^i}(\cdot)$ and $S_{\mathbf{a}^{i-1}}(\cdot)$. An algorithm is CA iff these intersections are on the envelop $S(\cdot)$ and $v_{i-1} \leq v_i \ \forall i \leq k$. The linear parametrization holds since the break point $v_i$ is linearly expressed in terms of the slopes and intercepts of the lines $S_{\mathbf{a}^i}(\cdot)$ and $S_{\mathbf{a}^{i-1}}(\cdot)$, while the intercepts are linear in the algorithm prices. Formally, this dependence is the product $Z_T(\boldsymbol{\gamma}^{\mathrm{B}}) J_T K_T(\boldsymbol{\gamma}^{\mathrm{B}}, \boldsymbol{\gamma}^{\mathrm{B}})$ of $k \times k$ matrices, where $J_T$ is a two-diagonal one with 1 on the diagonal and $-1$ under the diagonal; $Z_T(\boldsymbol{\gamma}) = \mathrm{diag}(z_1, \ldots, z_k)$, $z_j = (\boldsymbol{\gamma} \cdot \mathbf{a}^j - \boldsymbol{\gamma} \cdot \mathbf{a}^{j-1})^{-1}$ for $j = 1, \ldots, k$; and $K_T(\boldsymbol{\gamma}^{\mathrm{B}}, \boldsymbol{\gamma}') = ((\kappa_{ij}))_{i,j=1,\ldots,k}$, where $\kappa_{ij} = \gamma_t' a_t^i$ if the path $\mathbf{a}^i \in \mathfrak{S}_T$ passes through the node $\mathfrak{n}_j \in \mathfrak{N}_T$ whose round is $t = |\mathfrak{n}_j| + 1$, and $\kappa_{ij} = 0$, otherwise, for some fixed numbering of the nodes $\mathfrak{N}_T = \{\mathfrak{n}_j\}_{j=1}^k$[11]. All technical details are in App. A.2.2.

Finally, the parametrization via the break points $\{v_i\}_{i=1}^k$ allows to easily calculate the ESR of the algorithm. Indeed, the revenue $\mathrm{SRev}_{\boldsymbol{\gamma}^{\mathrm{S}}, \boldsymbol{\gamma}^{\mathrm{B}}}(\mathcal{A}, v)$ is constant on the intervals $(v_i, v_{i+1})$, because $\boldsymbol{\gamma}^{\mathrm{B}}$ is regular and the strategic buyer chooses only the strategy $\mathbf{a}^i$, when his valuation $v$ is in $(v_i, v_{i+1})$. Hence, the ESR is the sum of constant revenues on the intervals weighted by their probabilities: $\mathbb{E}[\mathrm{SRev}_{\boldsymbol{\gamma}^{\mathrm{S}}, \boldsymbol{\gamma}^{\mathrm{B}}}(\mathcal{A}, V)] = \sum_{i=1}^k (F_D(v_{i+1}) - F_D(v_i)) \mathrm{Rev}_{\boldsymbol{\gamma}^{\mathrm{S}}}(\mathcal{A}, \mathbf{a}^i)$, where $\mathrm{Rev}_{\boldsymbol{\gamma}^{\mathrm{S}}}(\mathcal{A}, \mathbf{a}^i)$ can be linearly expressed in terms of the algorithm prices and, thus, in terms of the break points $\{v_i\}_{i=1}^k$ (by means of our matrices introduced above). Integration by parts makes the ESR be a bilinear form of $\{1 - F_D(v_i)\}_{i=1}^k$ and $\{v_i\}_{i=1}^k$. We formalize it in the following proposition (the proof is in App. A.2.3), which implies Th. 3 since the class of CA algorithms $\tilde{\mathfrak{A}}_T$ contains an optimal pricing (by Prop. 1).

**Proposition 2.** *Let $T \in \mathbb{N}$, $\boldsymbol{\gamma}^{\mathrm{S}}$ be a discount, $\boldsymbol{\gamma}^{\mathrm{B}}$ be a regular discount, the strategies $\mathfrak{S}_T$ be naturally ordered by $\boldsymbol{\gamma}^{\mathrm{B}}$ and the matrix notations be introduced as above. Then there exists an invertible linear transformation $\mathbf{w}_{\boldsymbol{\gamma}^{\mathrm{B}}} : \tilde{\mathfrak{A}}_T(\boldsymbol{\gamma}^{\mathrm{B}}) \to \Delta^k, k = k(T)$, s.t., for any completely active pricing algorithm $\mathcal{A} \in \tilde{\mathfrak{A}}_T(\boldsymbol{\gamma}^{\mathrm{B}})$, its ESR has the form $\mathbb{E}_{V \sim D}[\mathrm{SRev}_{\boldsymbol{\gamma}^{\mathrm{S}}, \boldsymbol{\gamma}^{\mathrm{B}}}(\mathcal{A}, V)] = L_{D, \boldsymbol{\gamma}^{\mathrm{S}}, \boldsymbol{\gamma}^{\mathrm{B}}}(\mathbf{w}_{\boldsymbol{\gamma}^{\mathrm{B}}}(\mathcal{A}))$, where*

$$L_{D, \boldsymbol{\gamma}^{\mathrm{S}}, \boldsymbol{\gamma}^{\mathrm{B}}}(\mathbf{v}) := (1 - F_D(\mathbf{v}))^{\mathsf{T}} \Xi_T(\boldsymbol{\gamma}^{\mathrm{S}}, \boldsymbol{\gamma}^{\mathrm{B}}) \mathbf{v}, \ \mathbf{v} \in \Delta^k; \tag{4}$$

$\Xi_T(\boldsymbol{\gamma}^{\mathrm{S}}, \boldsymbol{\gamma}^{\mathrm{B}}) := J_T \cdot K_T(\boldsymbol{\gamma}^{\mathrm{B}}, \boldsymbol{\gamma}^{\mathrm{S}}) K_T(\boldsymbol{\gamma}^{\mathrm{B}}, \boldsymbol{\gamma}^{\mathrm{B}})^{-1} J_T^{-1} Z_T(\boldsymbol{\gamma}^{\mathrm{B}})^{-1}$ *is the invertible $k \times k$ matrix that depends only on the discounts; and the vector $(1 - F_D(\mathbf{v})) := \{1 - F_D(v_i)\}_{i=1}^k \in \mathbb{R}^k$.*

**Theorem 3.** *Let $T \in \mathbb{N}$ and $\gamma_{\mathrm{S}}, \gamma_{\mathrm{B}}$ be discount rates s.t. $\gamma_{\mathrm{S}} \geq \gamma_{\mathrm{B}}$ and the sequence $\boldsymbol{\gamma}^{\mathrm{B}} = \{\gamma_{\mathrm{B}}^{t-1}\}_{t=1}^T$ is regular. The optimization problem of finding an optimal algorithm is equivalent to maximization of the multivariate functional $L_{D, \boldsymbol{\gamma}^{\mathrm{S}}, \boldsymbol{\gamma}^{\mathrm{B}}}(\cdot)$ over the set $\Delta^k = \{\mathbf{v} \in \mathbb{R}^k | 0 \leq v_1 \leq \ldots \leq v_k\}, k = 2^T - 1$, i.e.,*

$$\max_{\mathcal{A} \in \mathfrak{A}_T} \mathbb{E}_{V \sim D}[\mathrm{SRev}_{\boldsymbol{\gamma}^{\mathrm{B}}, \boldsymbol{\gamma}^{\mathrm{S}}}(\mathcal{A}, V)] = \max_{\mathbf{v} \in \Delta^k} L_{D, \boldsymbol{\gamma}^{\mathrm{S}}, \boldsymbol{\gamma}^{\mathrm{B}}}(\mathbf{v}), \tag{5}$$

*where $L_{D, \boldsymbol{\gamma}^{\mathrm{S}}, \boldsymbol{\gamma}^{\mathrm{B}}}$ is defined in Eq. (4) and depends only on the discounts and the distribution $D$.*

It is quite important to emphasize that the $k$-dimensional functional $L_{D, \boldsymbol{\gamma}^{\mathrm{S}}, \boldsymbol{\gamma}^{\mathrm{B}}}$ is a *bilinear form* applied to the vectors $\mathbf{v}$ and $1 - F_D(\mathbf{v})$. This bilinear form is independent of the distribution $D$ and is defined by the matrix $\Xi_T(\boldsymbol{\gamma}^{\mathrm{S}}, \boldsymbol{\gamma}^{\mathrm{B}})$. In this view, there is a strong relationship between our optimization functional $L_{D, \boldsymbol{\gamma}^{\mathrm{S}}, \boldsymbol{\gamma}^{\mathrm{B}}}$ and the function $H_D$ (see Sec. 3): the functional $L_{D, \boldsymbol{\gamma}^{\mathrm{S}}, \boldsymbol{\gamma}^{\mathrm{B}}}$ constitutes the key basis of optimal algorithms in dynamic setting and is fundamental for them as the function $H_D(p) = p \mathbb{P}_{V \sim D}[V \geq p]$ is fundamental for optimal pricing in static auctions. Moreover, in the case of equal discounts $\gamma_{\mathrm{S}} = \gamma_{\mathrm{B}}$, the optimization of $L_{D, \boldsymbol{\gamma}^{\mathrm{B}}, \boldsymbol{\gamma}^{\mathrm{B}}}$ reduces to the maximization of $H_D$ (simple algebra is in App. A.2.4). Since, in the particular case of $\gamma_{\mathrm{S}} = \gamma_{\mathrm{B}}$, the optimization of $L_{D, \boldsymbol{\gamma}^{\mathrm{B}}, \boldsymbol{\gamma}^{\mathrm{B}}}$ has no closed form solution (it reduces to the optimization of $H_D$), we thus expect that, in the other cases, generally, our optimization problem does not admit a closed form solution as well.

In contrast to the initial form of our problem, *numerical optimization of the functional $L_{D, \boldsymbol{\gamma}^{\mathrm{B}}, \boldsymbol{\gamma}^{\mathrm{B}}}$ is much easier* (though it still has the same number of variables as the initial problem). First, the functional is continuously differentiable as many times as the CDF $F_D$. Second, its derivatives have simple form, $i, j = 1, \ldots, k$: $\partial_{v_i} L(\mathbf{v}) = -f_D(v_i) \sum_l \xi_{il} v_l + \sum_l (1 - F_D(v_l)) \xi_{li}, \ \partial_{v_i} \partial_{v_j} L(\mathbf{v}) =$

$-f_D(v_i)\xi_{ij} - f_D(v_j)\xi_{ji}$ for $i \neq j$, and $\partial^2_{v_i}L(\mathbf{v}) = -2f_D(v_i)\xi_{ii} - f'_D(v_i)\sum_l \xi_{il}v_l$, where $\xi_{ij}$ is the $ij$-th element of $\Xi_T(\boldsymbol{\gamma}^{\mathtt{S}}, \boldsymbol{\gamma}^{\mathtt{B}})$. The derivatives can be easily computed: see App. I for the pseudo-code that calculates $\xi_{ij}$. Third, the domain $\Delta^k$ is convex (moreover is closed when the support of $F_D$ is bounded) and has a simple form of simplex. Finally, the matrix $\Xi_T(\boldsymbol{\gamma}^{\mathtt{S}}, \boldsymbol{\gamma}^{\mathtt{B}})$ is positive definite on $\Delta^k$. Hence, a variety of gradient methods can be used to find the solution (see our experiments in Sec. 5).

**The step-by-step instruction to find the optimal pricing.** Remind that, for static pricing, the optimal (Myerson) price can be found from maximization of the functional $H_D(p) = p(1 - F_D(p))$. In our dynamic case, the optimal pricing algorithm can be found similarly as follows: (I) construct the matrix $\Xi$ (the pseudo-code to calculate its elements is in Appendix I); (II) construct the functional $L_{D,\boldsymbol{\gamma}^{\mathtt{B}},\boldsymbol{\gamma}^{\mathtt{B}}}(\cdot)$ from Eq. (4); (III) find a vector $\mathbf{v}^{\mathrm{Opt}}$ s.t. it maximizes $L_{D,\boldsymbol{\gamma}^{\mathtt{B}},\boldsymbol{\gamma}^{\mathtt{B}}}(\mathbf{v})$, e.g., an apply numerical method using derivatives of $L_{D,\boldsymbol{\gamma}^{\mathtt{B}},\boldsymbol{\gamma}^{\mathtt{B}}}(\cdot)$ provided in the previous paragraph; (IV) convert the vector $\mathbf{v}^{\mathrm{Opt}}$ to the prices of the optimal algorithm by means of the linear transformation $\mathbf{w}^{-1}_{\boldsymbol{\gamma}^{\mathtt{B}}}(\cdot)$, which is mentioned in Prop. 2 and whose matrix is $K_T(\boldsymbol{\gamma}^{\mathtt{B}}, \boldsymbol{\gamma}^{\mathtt{B}})^{-1}J_T^{-1}Z_T(\boldsymbol{\gamma}^{\mathtt{B}})^{-1}$ (see App. A.2.2).

**Remark 2.** In Appendix A.2, we show that all results of this section hold also for non-geometric discounts $\boldsymbol{\gamma}^{\mathtt{S}} = \{\gamma_t^{\mathtt{S}}\}_{t=1}^T$ and $\boldsymbol{\gamma}^{\mathtt{B}} = \{\gamma_t^{\mathtt{B}}\}_{t=1}^T$ such that $\gamma_{t+1}^{\mathtt{B}}/\gamma_t^{\mathtt{B}} \leq \gamma_{t+1}^{\mathtt{S}}/\gamma_t^{\mathtt{S}}$.

**Remark 3.** The regularity of the discount $\boldsymbol{\gamma}^{\mathtt{B}}$ is used to get: the uniqueness of $\boldsymbol{\gamma}$-dependent natural order of the strategies $\mathfrak{S}_T$ (for Prop. 2); zero probability of valuations for which the optimal buyer strategy is not unique (in Prop. 1). Ways to relax this restriction are discussed in App. D. In any way, non-regular discounts are rare, and do not affect our qualitative results in Sec. 5.

## 5 Efficient approximation, constrained optimization, numerical experiments

**Approximation by optimal $\tau$-step pricing ($\gamma_{\mathtt{S}} \geq \gamma_{\mathtt{B}}$).** In the case of infinite games, we have no similar powerful instrument to find an optimal pricing (unlike to the case of finite games in Sec. 4). Moreover, when the horizon $T$ is finite but sufficiently large, the optimization problem even in the simplified form of Eq. (5) suffers from dimensional complexity since the number of variables is $2^T - 1$. In both cases, however, we can approximate the optimal algorithm by an algorithm that is optimal in some finite dimensional subclass of $\mathfrak{A}_T, T \in \mathbb{N} \cup \{\infty\}$. Namely, for $\tau \in \mathbb{N}$, let us say that $\mathcal{A}$ is a $\tau$-*step pricing algorithm*, if $\forall \mathbf{a}, t > \tau : \mathcal{A}(\mathbf{a}_{1:t-1}) = \mathcal{A}(\mathbf{a}_{1:\tau-1})$, i.e., it is constant from the $\tau$-th round on. The set of all $\tau$-step algorithms is denoted by $\mathfrak{A}_T^\tau$. An attentive reader may note that the problem of finding an optimal $\tau$-step algorithm $\mathcal{A} \in \mathfrak{A}_T^\tau$ for the finite or infinite game is equivalent to finding an optimal algorithm for the $\tau$-round finite game with "shortened" discount sequences $\boldsymbol{\gamma}^{\mathtt{S},\tau} := (\gamma_1^{\mathtt{S}}, .., \gamma_{\tau-1}^{\mathtt{S}}, \sum_{t=\tau}^T \gamma_t^{\mathtt{S}})$ and $\boldsymbol{\gamma}^{\mathtt{B},\tau} := (\gamma_1^{\mathtt{B}}, .., \gamma_{\tau-1}^{\mathtt{B}}, \sum_{t=\tau}^T \gamma_t^{\mathtt{B}})$. Hence, one can apply the optimization technique from Th. 3 (which holds for $\boldsymbol{\gamma}^{\mathtt{B},\tau}$ and $\boldsymbol{\gamma}^{\mathtt{S},\tau}$ due to Remark 2). The following proposition (the proof is in App. A.3.1) formally states that the expected revenue of the optimal $\tau$-step algorithm $\mathcal{A}_\tau^* \in \mathfrak{A}_T^\tau$ converges to one of the optimal pricing $\mathcal{A}^* \in \mathfrak{A}_T$ when $\tau \to T$.

**Proposition 3.** *Let $T \in \mathbb{N} \cup \{\infty\}$ and $\boldsymbol{\gamma}^{\mathtt{S}}, \boldsymbol{\gamma}^{\mathtt{B}}$ be discount sequences s.t. $\gamma_{t+1}^{\mathtt{B}}/\gamma_t^{\mathtt{B}} \leq \gamma_{t+1}^{\mathtt{S}}/\gamma_t^{\mathtt{S}}, \Gamma_\tau^{\mathtt{S}} := \sum_{t=\tau+1}^T \gamma_t^{\mathtt{S}}$ for $\tau \in \mathbb{N}, \tau < T$. Then the following bounds hold:*

$$\max_{\mathcal{A} \in \mathfrak{A}_T^\tau} \mathbb{E}[\mathrm{SRev}_{\boldsymbol{\gamma}^{\mathtt{S}}, \boldsymbol{\gamma}^{\mathtt{B}}}(\mathcal{A}, V)] \leq \max_{\mathcal{A} \in \mathfrak{A}_T} \mathbb{E}[\mathrm{SRev}_{\boldsymbol{\gamma}^{\mathtt{S}}, \boldsymbol{\gamma}^{\mathtt{B}}}(\mathcal{A}, V)] \leq \max_{\mathcal{A} \in \mathfrak{A}_T^\tau} \mathbb{E}[\mathrm{SRev}_{\boldsymbol{\gamma}^{\mathtt{S}}, \boldsymbol{\gamma}^{\mathtt{B}}}(\mathcal{A}, V)] + \Gamma_\tau^{\mathtt{S}}\mathbb{E}[V]. \quad (6)$$

First, Prop. 3 provides the seller with *a tool to make a trade-off between the achievable fraction of the maximal revenue and the computational complexity of the optimization problem to be solved.* In particular, she is able to choose the parameter $\tau$ s.t. her computational capabilities on the dimension $2^\tau - 1$ of the optimization functional $L$ are fitted and the boost in the relative regret bound $\Gamma_\tau^{\mathtt{S}}\mathbb{E}[V]/\Gamma_1^{\mathtt{S}}\mathbb{E}[V] = \gamma_{\mathtt{S}}^{\tau-1}$ is minimal. Note that the seller can improve her revenue obtained from an optimal constant algorithm just by applying an optimal $\tau$-step algorithm for small $\tau$. For instance, for $\tau = 4$, this algorithm can be easily found in $2^\tau - 1 = 15$-dimensional space and provides noticeable boost in revenue (revenue improvement is illustrated in Fig. 1). Second, from Eq. (6), we have that the convergence bound is $\Gamma_\tau^{\mathtt{S}} = \gamma_{\mathtt{S}}^\tau/(1 - \gamma_{\mathtt{S}})$ and the convergence rate is $\Gamma_{\tau+1}^{\mathtt{S}}/\Gamma_\tau^{\mathtt{S}} = \gamma_{\mathtt{S}}$. On the one hand, it means that the smaller $\gamma_{\mathtt{S}}$ is, the faster the revenue of the suboptimal algorithm $\mathcal{A}_\tau^*$ converges to the optimal revenue, and, thus, *the functional $L$ in Eq. (4) with the smaller dimension should be optimized* to reach revenue close to the optimal one within $\epsilon$ error, $\epsilon > 0$[12]. On the other hand, the

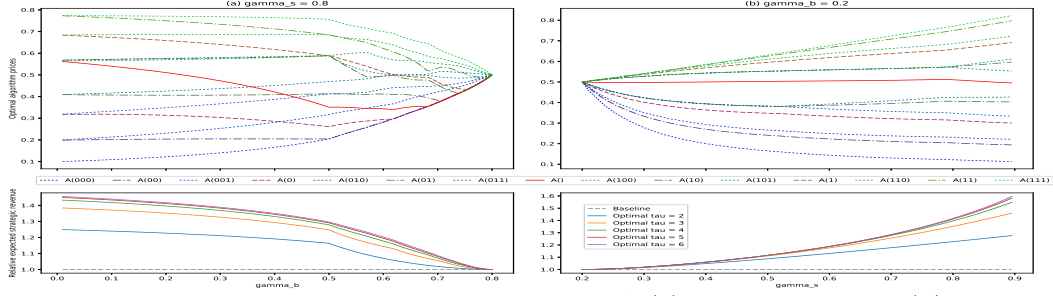

Figure 1: Infinite game $T = \infty$, uniform $D$. The prices $\mathcal{A}_4^*(\mathfrak{n})$, for nodes $\mathfrak{n} \in \mathfrak{N}$ s.t. $|\mathfrak{n}| \leq 3$, of the optimal 4-step algorithm $\mathcal{A}_4^*$ and the relative expected strategic revenue (w.r.t. $\mathcal{A}_1^*$) of the optimal $\tau$-step algorithm $\mathcal{A}_\tau^*$, $\tau = 2, .., 6$, for discounts: (a) $\gamma_S = 0.8$ and various $\gamma_B$; (b) $\gamma_B = 0.2$ and various $\gamma_S$.

slower convergence rate is, *the more revenue can be extracted from non-static pricing*. Namely, the closer $\gamma_S$ to 1 is, the larger the improvement of the revenue of the optimal $\tau$-step pricing is w.r.t. the constant pricing (for fixed $\gamma_B$ and $\tau$). This is both supported by our experiments (see growing relative revenue in Fig. 1(b, bottom) & Fig.C7 as $\gamma_S$ grows) and in line with the intuition: the larger $\gamma_S$ is, the more revenue could be earned in future rounds (and hence the more profitable dynamic pricing is).

**Optimal algorithms with constraints.** One more structural insight of our reduction in Sec. 4: *optimization over the set of break points $\{v_i\}$ of the surplus envelope $S$ allows to find optimal algorithms with constraints that can be expressed in terms of these break points*. In particular, the seller is able to control the probability of buyer usage of each strategy $\mathbf{a}^i \in \mathfrak{S}_T$ through a constraint on $F(v_{i+1}) - F(v_i)$ (e.g., setting it to zero). E.g., the seller is looking for an algorithm s.t. strategies active with positive probability are monotone, i.e. of the form $0^n 1^{T-n}$ for some $n \leq T$. Hence, if $\mathbf{a}^i$ is not monotone, then $v_i = v_{i+1}$, i.e. the line $S_{\mathbf{a}^i}$ is tangent to the envelope $S$ in only one point. To find an optimal algorithm among those for which $v_i = v_{i+1}$, one needs slightly update the functional $L$: replace $i$-th and $(i+1)$-th rows in the matrix $\Xi$ by their sum, do the same with $i$-th and $(i+1)$-th columns, and remove $i$-th components from the vectors $1 - F(\mathbf{v})$ and $\mathbf{v}$. The modified optimization functional for the problem with constraints will have $T+1$ variables since it is equal to the number of strategies that are active with positive probability. So, the dimensionality of the optimization problem can be reduced by means of constraints on the form of the algorithm, that can thus be find efficiently.

**Lower bound on the maximal revenue for $\gamma_S = 1$.** In this case, the algorithm PRRFES [30][Th.5] with optimal upper regret bound can be used to get a lower bound on the optimal ESR. Using PRRFES, the seller is able to increase her revenue w.r.t. the optimal constant pricing by up to $\mathbb{E}[V]/H_D(p_D^*) > 1$ (e.g., it is $+100\%$ when $D$ is uniform on $[0,1]$) as $T \to +\infty$. See details in App. G.

**Numerical experiments**[13]. To show the practical profit and properties of optimal algorithms obtained via our functional $L$ from Eq. (4) for the case $\gamma_S \geq \gamma_B$, we conducted numerical experiments in several representative games. We seek for optimal $\tau$-step algorithms $\mathcal{A}_\tau^*$, $\tau = 2, .., 6$, in infinite games with the valuation $V$ uniformly distributed in $[0,1]$[14], i.e., $F_D(v) = v$. Hence, the functional $L_{D,\gamma^S,\gamma^B}$ becomes thus quadratic and is optimized numerically using the Sequential Least Squares Programming. The ESR of the algorithms are compared with the expected revenue $H_D(p^*(D))\Gamma^S$ of the optimal constant pricing $\mathcal{A}_1^*$ (see Sec. 3), which is *treated as the baseline* from here on. Fig. 1 contains: the obtained in this way prices $\mathcal{A}_4^*(\mathfrak{n})$ for all nodes $\mathfrak{n}$ (at the top) and the relative expected strategic revenue of $\mathcal{A}_\tau^*$ (w.r.t. $\mathcal{A}_1^*$) for $\tau = 2, .., 6$ (at the bottom). The results in Fig. 1(a) are for $\gamma_S = 0.8$ and $\gamma_B \in \{0.01 + i \cdot 0.005\}_{i=0}^{148}$, while the ones in Fig. 1(b) for $\gamma_B = 0.2$ and $\gamma_S \in \{0.2 + i \cdot 0.005\}_{i=0}^{159}$.

First, at the bottom of Fig. 1, we see that *the optimal $\tau$-step algorithms $\mathcal{A}_\tau^*$ outperform the baseline optimal constant pricing $\mathcal{A}_1^*$ for any observed pair of discounts*. Moreover, Fig. 1 demonstrates that the significant increase in revenue can be obtained even when the minimal possible step aside from the constant pricing is made ($\tau = 2$). E.g., the seller can extract up to $+20\%$ revenue by just maximizing the functional Eq. (4) in the 3-dimensional space (since $2^\tau - 1 = 3$ for $\tau = 2$): e.g., the revenue improvement is larger than 20% for $\gamma_S = 0.9, \gamma_B = 0.2$, larger than 16% for $\gamma_S = 0.8, \gamma_B = 0.5$, and larger than 10% for $\gamma_S = 0.8, \gamma_B = 0.55$. Second, we see that the expected strategic revenue of $\mathcal{A}_\tau^*$ converges quite quickly to the optimal one (which thus larger than the revenue of the baseline $\mathcal{A}_1^*$ as well). This observation constitutes the empirical evidence of Prop. 3, which suggests that

the convergence rate is equal to $\gamma_S$. Third, the top part of Fig. 1 demonstrates us that *an optimal algorithm may be non-consistent*: e.g., the reverse order of the prices $\mathcal{A}_4^*(\mathfrak{e}) < \mathcal{A}_4^*(001)$ for $\gamma_B > \approx 0.57$ in Fig. 1(a). Fourth, if the distance between the discount rates $\gamma_S$ and $\gamma_B$ converges to 0, then the optimal algorithm $\mathcal{A}^*$ converges to the optimal constant one $\mathcal{A}_1^*$ (what experimentally supports that $H_D$ is a special case of $L_{D,\gamma^S,\gamma^B}$). More details and observations are in App. C.2.3. Overall, we conclude that *learning of prices even in several starting rounds allow to extract revenue significantly larger than the one of optimal static pricing*.

## 6   Incomplete information about buyer discount sequence

Our results can also be applied in the case of a *weak* assumption on the seller's information about the buyer's discount sequence. The weak assumption: the seller does not know the exact discount sequence of the buyer, but rather knows a set of intervals $\{[\gamma_t^0; \gamma_t^1]\}_{t=1}^T$ s.t. the discount coefficient $\gamma_t^B$ is located in $[\gamma_t^0, \gamma_t^1]$. We provide the interpretation of the model, which explains the foundation of the weak assumption. We also show the performance of our results adapted to the *weak* assumption setting. For the sake of exposition, all discount sequences are geometrical from here on in the section.

The discount in our model can be interpreted as the continuation probability, i.e., $\gamma$ is the probability that the game will continue for one more round. E.g., in the example from Sec. 1 (see App. F for an extended version as well), $\gamma$ is the probability that the user does not click on the ad and follows a link that is in the sight of the RTB platform. In this interpretation, the discount $\gamma$ is common. The difference in discounts appears, because the seller and the buyer do not know $\gamma$ exactly, but rather estimate it based on available information about the user. Let $\gamma = \gamma(\xi_1, \xi_2)$, where $\xi_1, \xi_2$ are user features. Assume that the seller observes both $\xi_1$ and $\xi_2$, while the buyer observes only $\xi_1$. Then the seller is able to estimate $\gamma$ accurately as well as to recover the buyer's estimate $\gamma_B(\xi_1)$. To sum up: it is likely that the seller in our model can at least recover the buyer discount $\gamma_B$ with high accuracy.

Let us consider two cases. Case (1): if the seller knows only a lower bound $\hat{\gamma}_B$ for $\gamma_B$ s.t. $\gamma_S < \hat{\gamma}_B$, then she can apply "Big deal", which prices are calculated using $\hat{\gamma}_B$: $\mathcal{A}_{bd}(\mathfrak{e}) = \sum_t \hat{\gamma}_B^{t-1} p_D^*$; $\mathcal{A}_{bd}(1 \circ \mathfrak{n}) = 0 \, \forall \mathfrak{n}$; $\mathcal{A}_{bd}(0 \circ \mathfrak{n}) = T p_D^* \, \forall \mathfrak{n}$. Buyer (whose discount $\gamma_B \geq \hat{\gamma}_B$) with valuation $v > p_D^*$ still accepts the first proposed price, hence, the seller gets at least $\sum_t \hat{\gamma}_B^{t-1} p_D^* (1 - F(p_D^*))$. This is less than the optimal revenue (when $\gamma_B$ is known exactly), but strictly larger than the one of static pricing. Similarly, modifications of "Big deal" can be applied when seller knows only distribution of $\gamma_B, \gamma_B \geq \gamma_S$. Case (2): The seller uses the functional $L$ to find an optimal algorithm, assumes buyer's discount is $\gamma_B' = \gamma_B + \varepsilon$, but faces a buyer with true discount $\gamma_B$. We evaluate the loss in revenue by the following numerical experimentation: $T = 5$, $V \sim U[0;1]$ (uniform on $[0;1]$) and $\gamma_S = 0.5$ (different sets of parameters give qualitatively the same results). In figure above, the expected strategic revenue (ESR) of this seller is divided by the ESR of a well-informed seller (i.e. s.t. $\varepsilon = 0$). We see: (a) if $\varepsilon$ is small enough (for $\varepsilon = 0.02$, or $\geq 4\%$ of $\gamma_B$), then S still able to extract over 99% of the optimal ESR; (b) even if $\varepsilon$ is very large (for $\varepsilon = 0.1$, or $\geq 20\%$ of $\gamma_B$) S still able to extract over 97% of the optimal ESR for most cases ($\gamma_B \leq 0.4$); and

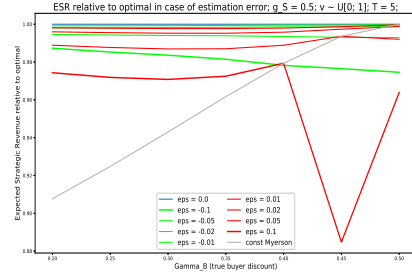

(c) if S is able to just separate $\gamma_B$ of $\gamma_S$ with a decent margin, then she is able to gain extra revenue.

## 7   Conclusions

We studied online learning algorithms that maximize expected cumulative revenue of repeated posted-price auctions with a strategic buyer that holds a fixed private valuation. First, when the participants non-equally discount their cumulative utilities, we showed that the constant pricing, surprisingly, is no longer optimal. Second, for the case of more patient seller, we introduced a novel multidimensional optimization functional which is a multivariate analogue of the one used to determine Myerson's price. This functional can be used (1) to find an optimal dynamic pricing, i.e., by efficient gradient-based methods; and (2) to construct an optimal $\tau$-step algorithm (low-dimensional approximation) that allows the seller to improve her revenue even in the game with a large horizon $T$. Finally, we conducted extensive numerical analysis to show that optimal algorithms are non-trivial, may be non-consistent, and generate larger expected revenue than the constant pricing with Myerson's price.

## Footnotes

[1]There were works (e.g., [7, 8, 30, 32, 33]), where only buyer utilities were discounted ,while the seller's ones did not. But, those studies considered worst-case regret optimization, which is different from our setting.

[2]This pricing offers an up-front payment for all copies of a good for Myerson's price in the first round.

[3]A $\tau$-step algorithm plays all the $T$ rounds, but its prices do not change after the round $\tau < T$.

[4]Our results still can be applied in the case when the seller possesses only incomplete information about the buyer's discount sequence. See 6 for more details.

[5]Alternatively, a discount factor can model the patience level of a participant to wait for instant revenue [7, 30].

[6]RTB platforms run $10^8$ auctions a day: commitment violation will be easily seen by advertisers, see App.F.

[7]This price can be find by the equation $p = (1 - F_D(p))/f_D(p)$, when $D$ has continuous probability density $f_D$.

[8]A similar pricing was in [49] for mechanism environments with multiplicative separability.

[9]Moreover, for $T = \infty$, this revenue improvement is $\Gamma^B / \Gamma^S = \frac{1-\gamma_S}{1-\gamma_B}$ and goes to $+\infty$ as $\gamma_B \to 1-$ for a fixed $\gamma_S$.

[10]In a piece (an interval $(v_i, v_{i+1})$) the function $S(\cdot)$ equals to the function $S_{\mathbf{a}^i}(\cdot)$ for some strategy $\mathbf{a}^i$ which is a linear function of $v$: $S_{\mathbf{a}^i}(v) = (\sum_t \gamma_t^{\mathrm{B}} a_t^i) v - (\sum_t \gamma_t^{\mathrm{B}} a_t^i p_t)$, see Def. 2.

[11]Note: by the definition, the $i$-th component of the vector $K_T(\boldsymbol{\gamma}^{\mathrm{B}}, \boldsymbol{\gamma}') \mathcal{A}$ is equal to $\sum_{t=1}^T \gamma_t' a_t^i \mathcal{A}(a_1^i \ldots a_{t-1}^i)$.

[12]Take $\tau > \tau_{\gamma_{\mathtt{S}}, D, \epsilon} := \log_{\gamma_{\mathtt{S}}}(\epsilon(1-\gamma_{\mathtt{S}})\mathbb{E}[V])$ to be $\epsilon$-close to the optimal revenue. Note that $\tau_{\gamma_{\mathtt{S}}, D, \epsilon} \to_{\gamma_{\mathtt{S}} \to 0} 0$.

[13]The code of all our experiments is avail. at `https://github.com/theonlybars/neurips-2019-rppa`.

[14]Experiments for other distributions and horizons are presented in App. C. The results for them are similar.

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
