[Supplementary Material]

# Optimal Pricing in Repeated Posted-Price Auctions with Different Patience of the Seller and the Buyer: SUPPLEMENTARY MATERIALS

### Arsenii Vanunts[*]      Alexey Drutsa[†]

## Contents

[*]Yandex, Moscow, Russia (www.yandex.com); e-mail: `avanunts@yandex.ru`.

[†]Yandex, Moscow, Russia (www.yandex.com); Faculty of Mechanics and Mathematics, Lomonosov Moscow State University, Moscow, Russia; e-mail: `adrutsa@yandex.ru`.

# A  Missing proofs

## A.1  Missed Proofs from Section 3

**Remark A.1.** All the proofs are provided for arbitrary discounts that satisfy $\boldsymbol{\gamma}^{\mathsf{S}} \leq \boldsymbol{\gamma}^{\mathsf{B}}$. Hence, they hold also for geometrical discounts with $\gamma_{\mathsf{S}} \leq \gamma_{\mathsf{B}}$, because they are particular cases of discounts that satisfy $\boldsymbol{\gamma}^{\mathsf{S}} \leq \boldsymbol{\gamma}^{\mathsf{B}}$.

**Remark A.2.** Note that both an optimal buyer strategy and an optimal algorithm will remain optimal, if the discount $\boldsymbol{\gamma}^{\mathsf{B}}$ or $\boldsymbol{\gamma}^{\mathsf{S}}$ is multiplied by any positive constant. Hence, from here on we assume w.l.o.g. that $\gamma_1^{\mathsf{B}} = 1$ and $\gamma_1^{\mathsf{S}} = 1$.

### A.1.1  Optimality of the constant Myerson pricing via a feasible mechanism when $\gamma_{\mathsf{S}} = \gamma_{\mathsf{B}}$

One can apply the almost folklore technique of reducing the case of equal discounts to a single-round feasible mechanism [1]. For any algorithm $\mathcal{A} \in \mathfrak{A}_T$, one constructs the feasible mechanism[1] $\mathbf{M}_{\mathcal{A}}$ which proceeds as follows: after the buyer's report of his valuation $v$, the mechanisms $\mathbf{M}_{\mathcal{A}}$ chooses a random round ($s$-th round is chosen with the probability proportional to the discount coefficient: $\gamma^{s-1}/\Gamma$) and simulates it according to the optimal buyer strategy $\mathbf{a}^{\mathrm{Opt}}(\mathcal{A}, v, \boldsymbol{\gamma}) = \{a_t^o(v)\}_{t=1}^T$, i.e., allocates the good for the price $\mathcal{A}(a_1^o(v) \dots a_{s-1}^o(v))$, if $a_s^o(v) = 1$.

**Proposition A.1.** *The mechanism $\mathbf{M}_{\mathcal{A}}$ is feasible (i.e., it satisfies correctness $\mathbf{Q}_{\mathcal{A}}(v) \leq 1$, incentive-compatibleness and individual rationality conditions; see [8, Sec.3]).*

*Proof.* Formally the allocation $\mathbf{Q}_{\mathcal{A}} : [0; +\infty) \to [0, 1]$ and the payment $\mathbf{P}_{\mathcal{A}} : [0; +\infty) \to [0; +\infty)$ of the feasible mechanism $\mathbf{M}_{\mathcal{A}} = (\mathbf{Q}_{\mathcal{A}}, \mathbf{P}_{\mathcal{A}})$ defined above are the following:

$$
\begin{aligned}
\mathbf{P}_{\mathcal{A}}(v) &:= \frac{\sum_{t=1}^T a_t^o(v) \gamma^{t-1} \mathcal{A}\big(a_1^o(v) \dots a_{t-1}^o(v)\big)}{\Gamma} \stackrel{\text{def}}{=\!=} \frac{\mathrm{SRev}_{\gamma,\gamma}(\mathcal{A}, v)}{\Gamma}, \\
\mathbf{Q}_{\mathcal{A}}(v) &:= \frac{\sum_{t=1}^T a_t^o(v) \gamma^{t-1}}{\Gamma}.
\end{aligned}
\tag{A.1}
$$

Let us denote $\mathrm{SSur}_{\gamma}(\mathcal{A}, v) := \mathrm{Sur}_{\gamma}(\mathcal{A}, v, \mathbf{a}^{\mathrm{Opt}}(\mathcal{A}, v, \boldsymbol{\gamma}))$.

The correctness $\mathbf{Q}_{\mathcal{A}}(v) \leq 1$ is trivially satisfied.

The individual-rationality condition is that the buyer expect to gain non-negative utility: $\forall v \geq 0 : \mathbf{Q}_{\mathcal{A}}(v) \cdot v - \mathbf{P}_{\mathcal{A}}(v) \geq 0$. Note that the latter term of the left-hand side is the normalized strategic surplus $\Gamma^{-1} \mathrm{SSur}_{\gamma}(\mathcal{A}, v)$ which is not less than the normalized surplus $\Gamma^{-1} \mathrm{Sur}_{\gamma}(\mathcal{A}, v, \mathbf{0}^T) = 0$ for the strategy $\mathbf{0}^T$ (the strategy that rejects all offered prices), by the definition of $\mathbf{a}^{\mathrm{Opt}}$. Thus, the individual-rationality condition holds.

Finally, the incentive compatibleness is that $\forall v, u \geq 0 : \mathbf{Q}_{\mathcal{A}}(v) \cdot v - \mathbf{P}_{\mathcal{A}}(v) \geq \mathbf{Q}_{\mathcal{A}}(u) \cdot v - \mathbf{P}_{\mathcal{A}}(u)$ due to that the right-hand side of the inequality is the expression $\mathrm{Sur}_{\gamma}(\mathcal{A}, v, \mathbf{a}^{\mathrm{Opt}}(\mathcal{A}, u, \boldsymbol{\gamma}))$ (again, by the definition of $\mathbf{a}^{\mathrm{Opt}}$). Thus, the mechanism $\mathbf{M}_{\mathcal{A}}$ is indeed feasible. $\square$

The expected revenue of $\mathbf{M}_{\mathcal{A}}$ is $\Gamma^{-1} \mathbb{E}\left[\mathrm{SRev}_{\gamma,\gamma}(\mathcal{A}, V)\right]$, where $\Gamma := \sum_{t=1}^T \gamma^{t-1}$. From [8], we know that the expected revenue of any feasible mechanism for the considered environment (single object, one agent) cannot be greater than $p_D^*(1 - F_D(p_D^*))$, where $F_D$ is the CDF of our valuation variable $V \sim D$ and $p_D^*$ is the *Myeson price*, i.e., the price that maximizes the functional $H_D(p) :=$

$p\mathbb{P}[V \geq p] = p(1 - F_D(p))^2$. Therefore, bringing these together, we obtain the following upper bound:

$$\mathbb{E}\left[\text{SRev}_{\gamma,\gamma}(\mathcal{A}, V)\right] \leq \Gamma p^*(1 - F_D(p^*)) \quad \forall \mathcal{A} \in \mathfrak{A}_T. \tag{A.2}$$

### A.1.2 Detailed proof of "big deal" optimality

Recall the "big deal" algorithm, which is defined in Sec.3. Here we prove its optimality in details.

**Proposition A.2.** *Let* $\boldsymbol{\gamma}^{\mathsf{S}}, \boldsymbol{\gamma}^{\mathsf{B}}$ *be discounts. The ESR of* $\mathcal{A}_{\mathsf{bd}}$ *is equal to* $\Gamma^{\mathsf{B}} H_D(p_D^*)$.

*Proof.* First, note that the buyer has no incentive to lie after the first round since the algorithm prices $p_t, t \geq 2$, do not depend on his decisions $a_t, t \geq 2$. Hence, possible candidates for optimal strategies are $0^\infty$, $1^\infty$, $01^\infty$, and $10^\infty$. It easy to see that the optimal buyer strategy in response to $\mathcal{A}_{\mathsf{bd}}$ is $1^\infty$ for the case $v > p_D^*$ and $0^\infty$ for $v < p_D^*$. Indeed, if the buyer accepts $p_1 := \mathcal{A}_{\mathsf{bd}}(\mathfrak{e}) = \Gamma^{\mathsf{B}} p_D^*$, further offers are for free goods that will be accepted. If the buyer rejects $p_1$, then, for any strategy $\mathbf{a} \in \mathfrak{S}_T$ s.t. $a_1 = 0$, we have

$$S_{\mathbf{a}}(v) \leq (\Gamma^{\mathsf{B}} - 1)(v - p^*) < \Gamma^{\mathsf{B}}(v - p^*) = S_{1^\infty}(v). \tag{A.3}$$

Thus, if $S_{1^\infty}(v) > 0 = S_{0^\infty}(v)$, then $1^\infty$ is optimal strategy, and, if $S_{1^\infty}(v) < 0$, then Eq. (A.3) implies optimality of $0^\infty$. Hence, the expected strategic revenue of $\mathcal{A}_{\mathsf{bd}}$ is

$$\mathbb{E}\left[\text{SRev}_{\boldsymbol{\gamma}^{\mathsf{S}}, \boldsymbol{\gamma}^{\mathsf{B}}}(\mathcal{A}_{\mathsf{bd}}, V)\right] = \mathbb{P}[p_D^* \leq V] \cdot \Gamma^{\mathsf{B}} \cdot p_D^* = H_D(p_D^*)\Gamma^{\mathsf{B}} = \mathbb{E}\left[\text{SRev}_{\boldsymbol{\gamma}^{\mathsf{B}}, \boldsymbol{\gamma}^{\mathsf{B}}}(\mathcal{A}_1^*, V)\right]. \tag{A.4}$$

$\square$

Bringing Proposition A.2 together with the bound Eq.(2), obtained in Sec.3, we obtain the optimality of $\mathcal{A}_{\mathsf{bd}}$ for the case $\boldsymbol{\gamma}^{\mathsf{S}} \leq \boldsymbol{\gamma}^{\mathsf{B}}$. Note that all the above text applies to arbitrary discounts, not just the geometric ones.

## A.2 Missed Proofs from Section 4

**Definition A.1.** For a discount sequence $\boldsymbol{\gamma} = \{\gamma_t\}_{t=1}^T$, we define *the discount rate* sequence $\boldsymbol{\nu}(\boldsymbol{\gamma}) := \{\nu_t(\boldsymbol{\gamma})\}_{t=1}^{T-1}$ as the sequence of the ratios of consecutive components of $\boldsymbol{\gamma}$: $\nu_t(\boldsymbol{\gamma}) := \gamma_{t+1}/\gamma_t$

**Remark A.3.** Let $\boldsymbol{\gamma}^1 = \{\gamma_t^1\}_{t=1}^T$ and $\boldsymbol{\gamma}^2 = \{\gamma_t^2\}_{t=1}^T$ be some discounts sequences. Then, the condition $\boldsymbol{\nu}(\boldsymbol{\gamma}^2) \geq \boldsymbol{\nu}(\boldsymbol{\gamma}^1)$ is equivalent to the one that the sequence $\{\gamma_t^2/\gamma_t^1\}_{t=1}^T$ is non-decreasing. The proof of this statement straightforwardly follows from Definition A.1.

**Remark A.4.** All the proofs are provided for arbitrary discounts that satisfy $\boldsymbol{\nu}(\boldsymbol{\gamma}^{\mathsf{S}}) \geq \boldsymbol{\nu}(\boldsymbol{\gamma}^{\mathsf{B}})$. Hence, they hold also for geometrical discounts with $\gamma_{\mathsf{S}} \geq \gamma_{\mathsf{B}}$, because they are particular cases of discounts that satisfy $\boldsymbol{\nu}(\boldsymbol{\gamma}^{\mathsf{S}}) \geq \boldsymbol{\nu}(\boldsymbol{\gamma}^{\mathsf{B}})$.

**Remark A.5.** Note that both an optimal buyer strategy and an optimal algorithm will remain optimal, if the discount $\boldsymbol{\gamma}^{\mathsf{B}}$ or $\boldsymbol{\gamma}^{\mathsf{S}}$ is multiplied by any positive constant. Hence, from here on in our paper we assume w.l.o.g. that $\gamma_1^{\mathsf{B}} = 1$ and $\gamma_1^{\mathsf{S}} = 1$.

### A.2.1 Full proof of Proposition 1

The right and left children of a node $\mathfrak{n}$ are denoted by $\mathfrak{r}(\mathfrak{n})$ and $\mathfrak{l}(\mathfrak{n})$ respectively. The left (right) subtrees rooted at the node $\mathfrak{l}(\mathfrak{n})$ ($\mathfrak{r}(\mathfrak{n})$ resp.) are denoted by $\mathfrak{L}(\mathfrak{n})$ ($\mathfrak{R}(\mathfrak{n})$ resp.). The operators $\mathfrak{l}(\cdot)$ and $\mathfrak{r}(\cdot)$ sequentially applied $s$ times to a node $\mathfrak{n}$ are denoted by $\mathfrak{l}^s(\mathfrak{n})$ and $\mathfrak{r}^s(\mathfrak{n})$ respectively, $s \in \mathbb{N}$.

*Proof of Proposition 1.* For a given algorithm and a given discount $\gamma^{\mathsf{S}}$, we will use the notation $r_{\mathbf{a}} := \mathrm{Rev}_{\gamma^{\mathsf{S}}}(\mathcal{A}, \mathbf{a})$ for any $\mathbf{a} \in \mathfrak{S}_T$ (similarly to Remark B.1, but indicating explicitly the seller's discount). The main idea of the proof consists in the following technique. We will consider all strategies $\mathbf{a}$ s.t. $S_{\mathbf{a}}(v) < S(v) \;\; \forall v \in [0; +\infty)$ (referred to as *non-active*), and, consequently, for each of them denoted by $\mathbf{a}$, we apply the following procedure of modifying the source algorithm $\mathcal{A}$: define a transformation $\mathcal{A}'$ that does not change $S_{\mathbf{b}}$ for $\mathbf{b} \in \mathfrak{S}_T \setminus \{\mathbf{a}\}$, moves $S_{\mathbf{a}}$ to the left until it is tangent to $S$ in some $v \in [0; +\infty)$, decreases $r_{\mathbf{a}}$, and does not decrease $r_{\mathbf{b}}$ for $\mathbf{b} \in \mathfrak{S}_T \setminus \{\mathbf{a}\}$. That will imply that the expected strategic revenue of the transformed algorithm $\mathcal{A}'$ is no lower than the one of the source algorithm $\mathcal{A}$. In this way, we will (one-by-one) make all strategies active.

Let us consider the set of all non-active strategies. If it is empty, then $\mathcal{A} \in \tilde{\mathfrak{A}}_T(\gamma^{\mathsf{B}})$ and Eq. (3) from Sec. 4 holds. Otherwise, note that the "always-reject" strategy $\mathbf{a} = 0^T$ is always active, since $S_{\mathbf{a}}(0) = 0 = S(0)$. Hence, one can order all non-active strategies by "the last $\mathbf{1}$ index" $t_1(\mathbf{a}) = \max\{t \mid a_t = 1\}$.

We take a non-active strategy $\mathbf{a}$ with the smallest $t_1(\mathbf{a})$, denoting $t_1 := t_1(\mathbf{a})$ and the node $\mathfrak{n} := a_1 a_2 \ldots a_{t_1-1}$, and construct a new algorithm $\mathcal{A}'$ based on the source one $\mathcal{A}$ in the following way. Set $\mathcal{A}' = \mathcal{A}$ and transform the prices $\mathcal{A}'(\mathfrak{n}), \mathcal{A}'(\mathfrak{r}(\mathfrak{n})), \ldots, \mathcal{A}'(\mathfrak{l}^{T-t_1-1}(\mathfrak{r}(\mathfrak{n})))$ as follows:

1. decrease $\mathcal{A}'(\mathfrak{n})$ until the function $S_{\mathbf{a}}$ is tangent to the function $S$ in some $v \in [0; +\infty)$;

2. if $t_1 < T$, increase $\mathcal{A}'(\mathfrak{l}^j(\mathfrak{r}(\mathfrak{n})))$ for $j = 0, \ldots, T - t_1 - 1$ in such a way that

$$\gamma^{\mathsf{B}}_{t_1} \cdot \mathcal{A}'(\mathfrak{n}) + \gamma^{\mathsf{B}}_{t_1+j+1} \cdot \mathcal{A}'(\mathfrak{l}^j(\mathfrak{r}(\mathfrak{n}))) = \mathrm{const}. \tag{A.5}$$

Since we chosen $\mathbf{a}$ with the smallest $t_1(\mathbf{a})$ among non-active strategies the price $\mathcal{A}'(\mathfrak{n})$ obtained in the step 1 is non-negative (and, thus, this step is correct). Indeed, substitute the $t_1$-th component in $\mathbf{a}$ by 0 and denote the obtained strategy by $\mathbf{b}$. Due to selection of $\mathbf{a}$, the strategy $\mathbf{b}$ is active. Therefore, assume $\mathcal{A}'(\mathfrak{n})$ is decreased to 0, then the function $S_{\mathbf{a}}(v)$ becomes equal to $S_{\mathbf{b}}(v) + \gamma^{\mathsf{B}}_{t_1} v$ by the definition. Since $S_{\mathbf{b}}$ is tangent to $S$, the increase of its slope by $\gamma^{\mathsf{B}}_{t_1}$ will result in intersection with $S$. This means that $S_{\mathbf{a}}$ will be tangent to $S$ before $\mathcal{A}'(\mathfrak{n})$ reaches 0.

In Fig.A.1, we visually show how our tuning technique works for an example.

Now let us prove that the transformation $\mathcal{A}'$ satisfies properties announced at the beginning of the proof. Let $\mathbf{b} \in \mathfrak{S}_T \setminus \{\mathbf{a}\}$. The step 2 implies that the transformation does not change $S_{\mathbf{b}}$. For a strategy $\mathbf{b}$ that does not come through the node $\mathfrak{r}(\mathfrak{n})$, the revenue $r_{\mathbf{b}}$ remains the same, since the algorithm prices that contribute to $r_{\mathbf{b}}$ are not altered. For $\mathbf{b} \neq \mathbf{a}$ that comes through the node $\mathfrak{r}(\mathfrak{n})$, let us prove that $r_{\mathbf{b}}$ can only increase. Since $\mathbf{b} \neq \mathbf{a}$ there is a round $t = t_1 + j + 1, j \geq 0$, where $b_t = 1$. Let $j$ s.t. this $t$ is the first round of acceptance after reaching the node $\mathfrak{r}(\mathfrak{n})$, and let us denote the node where this acceptance take place by $\mathfrak{m} := \mathfrak{l}^j(\mathfrak{r}(\mathfrak{n}))$. Therefore, one can write the following expression for the increment of $r_{\mathbf{b}}$: $\gamma^{\mathsf{S}}_{t_1} \left( \mathcal{A}'(\mathfrak{n}) - \mathcal{A}(\mathfrak{n}) + (\gamma^{\mathsf{S}}_{t_1+j+1}/\gamma^{\mathsf{S}}_{t_1})(\mathcal{A}'(\mathfrak{m}) - \mathcal{A}(\mathfrak{m})) \right) = = \gamma^{\mathsf{S}}_{t_1} \left( -(\gamma^{\mathsf{B}}_{t_1+j+1}/\gamma^{\mathsf{B}}_{t_1})(\mathcal{A}'(\mathfrak{m}) - \mathcal{A}(\mathfrak{m})) + (\gamma^{\mathsf{S}}_{t_1+j+1}/\gamma^{\mathsf{S}}_{t_1})(\mathcal{A}'(\mathfrak{m}) - \mathcal{A}(\mathfrak{m})) \right) \geq 0$, where we used Eq. (A.5) to obtain the first equation and used $\nu(\gamma^{\mathsf{B}}) \leq \nu(\gamma^{\mathsf{S}})$ to obtain the last inequality. So, $r_{\mathbf{b}}$ can only increase for $\mathbf{b} \in \mathfrak{S}_T \setminus \{\mathbf{a}\}$.

Finally, since $S_{\mathbf{a}}$ becomes tangent to $S$, which is convex (see Remark B.1), the function $S_{\mathbf{a}}$ either equals to $S$ exactly in one point $v \in [0; +\infty)$ or coincides with $S_{\mathbf{b}}$ for some $\mathbf{b} \in \mathfrak{S}_T \setminus \{\mathbf{a}\}$. The

**Figure A.1:** A visual example of how our tuning technique from the proof of Proposition 1 works. Here $T = 2, \gamma_{\mathsf{S}} = 3.0 > \gamma_{\mathsf{B}} = 0.7$. Initial algorithm is not active: its prices are $\mathcal{A}(\mathfrak{e}) = 0.5, \mathcal{A}(0) = 0.2, \mathcal{A}(1) = 0.6$. Tuned algorithm has prices $\hat{\mathcal{A}}(\mathfrak{e}) = 0.374, \hat{\mathcal{A}}(0) = 0.2, \hat{\mathcal{A}}(1) = 0.78$ and is active.

latter case is impossible since a function $S_{\mathbf{b}}$ have different slope for different strategy $\mathbf{b}$, because of regularity of $\gamma^{\mathsf{B}}$. Therefore, the optimal strategy does not change for the buyer with any valuation $v$ except the only one s.t. $S_{\mathbf{a}}(v) = S(v)$, and the strategic revenue expectation is not affected by the decrease of $r_{\mathbf{a}}$ (due to continuity of the valuation distribution $D$). Thus, $\mathbb{E}\left[\mathrm{SRev}_{\gamma^{\mathsf{S}}, \gamma^{\mathsf{B}}}(\mathcal{A}, V)\right] \leq \mathbb{E}\left[\mathrm{SRev}_{\gamma^{\mathsf{S}}, \gamma^{\mathsf{B}}}(\mathcal{A}', V)\right]$ and the number of non-active strategies of $\mathcal{A}'$ is reduced by one w.r.t. $\mathcal{A}$. After that, we repeatedly apply the above described transformation to $\mathcal{A}'$ until the resulted algorithm has no non-active strategies. In this way, we get $\tilde{\mathcal{A}} \in \tilde{\mathfrak{A}}_T$ that satisfies Eq. (A.5). $\qquad\square$

### A.2.2 Proof of linear mapping of CA algorithms onto $\Delta^k$ (Lemma A.1)

Let us remind the notations.

Our goal is to show that this class of algorithms $\tilde{\mathfrak{A}}_T$ can be linearly parametrized by the set $\Delta^k := \{\mathbf{v} = \{v_j\}_{j=1}^k \in \mathbb{R}^k \mid 0 \leq v_1 \leq \cdots \leq v_k\}$, where $k := k(T) := 2^T - 1$. In order to do this, first of all, we introduce several matrix and vector notations. First, from here on in our paper we fix an order of nodes $\mathfrak{N}_T = \{\mathfrak{n}_1, \ldots, \mathfrak{n}_k\}$[3], and, given this, we represent an algorithm $\mathcal{A} \in \mathfrak{A}_T$ as the vector

of its prices $\mathcal{A} = (\mathcal{A}(\mathfrak{n}_1), \ldots, \mathcal{A}(\mathfrak{n}_k))$; note we use the same notation both for the algorithm and its vector representation, since the object type could be easily restored from the context where it is used. We also introduce the map $\mathbf{p} : \mathfrak{S}_T \times \mathfrak{A}_T \to \mathbb{R}^T$, where $\mathbf{p}(\mathbf{a}, \mathcal{A})$ is the vector of consecutively offered prices by the algorithm $\mathcal{A} \in \mathfrak{A}_T$ along the path $\mathbf{a} \in \mathfrak{S}_T$.

Second, given a regular discount $\boldsymbol{\gamma}$, we introduce the notion of $\boldsymbol{\gamma}$-*dependent natural order* of the buyer strategies $\mathfrak{S}_T = \{0, 1\}^T$: $\mathbf{a} \prec_{\boldsymbol{\gamma}} \mathbf{b} \Leftrightarrow \boldsymbol{\gamma} \cdot \mathbf{a} < \boldsymbol{\gamma} \cdot \mathbf{b}$ for any $\mathbf{a}, \mathbf{b} \in \mathfrak{S}_T$. The important property of this order consists in that the slope of the $\boldsymbol{\gamma}$-discounted surplus function $S_{\mathbf{a}}$ is lower than the one of $S_{\mathbf{b}}$ when $\mathbf{a} \prec_{\boldsymbol{\gamma}} \mathbf{b}$. Using this order, we index the strategies: $\mathfrak{S}_T = \{\mathbf{a}^0, \ldots, \mathbf{a}^k\}$; note that the strategy $0^T$ is always the first one $\mathbf{a}^0$, while the strategy $1^T$ is the last one $\mathbf{a}^k$. Third, given another discount $\boldsymbol{\gamma}'$, we introduce the payment vector $\mathbf{r}(\boldsymbol{\gamma}', \boldsymbol{\gamma}, \mathcal{A})$, whose $j$-th component is $r_j(\boldsymbol{\gamma}', \boldsymbol{\gamma}, \mathcal{A}) := \boldsymbol{\gamma}' \cdot \mathbf{p}(\mathbf{a}^j, \mathcal{A})$ for $j = 1, \ldots, k$ (note that we exclude the zero payment corresponded to the zeroth strategy $\mathbf{a}^0$). We treat all vectors as vector-columns in our matrix operations.

Finally, we introduce the following $k \times k$ matrices:

- $J_T$ is a two-diagonal matrix with 1 on the diagonal and $-1$ under the diagonal;

- $Z_T(\boldsymbol{\gamma}) = \mathrm{diag}(z_1, \ldots, z_k)$, where with $z_j = (\boldsymbol{\gamma} \cdot \mathbf{a}^j - \boldsymbol{\gamma} \cdot \mathbf{a}^{j-1})^{-1}$ for $j = 1, \ldots, k$;

- $K_T(\boldsymbol{\gamma}, \boldsymbol{\gamma}') = ((\kappa_{ij}))_{i,j=1,\ldots,k}$, where $\kappa_{ij} = \gamma'_t a^i_t$ if the path $\mathbf{a}^i \in \mathfrak{S}_T$ passes through the node $\mathfrak{n}_j \in \mathfrak{N}_T$ whose round is $t$[4], and $\kappa = 0$, otherwise. Note that, by the definition, the $i$-th component of the vector $K_T(\boldsymbol{\gamma}, \boldsymbol{\gamma}')\mathcal{A}$ is equal to $\sum_{t=1}^{T} \gamma'_t a^i_t \mathcal{A}(a^i_1 \ldots a^i_{t-1})$.

**Lemma A.1.** *Let* $T \in \mathbb{N}, \boldsymbol{\gamma}$ *be a regular discount, the strategies* $\mathfrak{S}_T$ *be naturally ordered by* $\boldsymbol{\gamma}$ *(as above) and the matrix and vector notations be introduced as above. Then the set of CA (for* $\boldsymbol{\gamma}$*) algorithms* $\tilde{\mathfrak{A}}_T(\boldsymbol{\gamma})$ *(i.e., their vector representations) can be linearly mapped onto* $\Delta^{k(T)}$ *by the matrix* $W_T(\boldsymbol{\gamma}) := Z_T(\boldsymbol{\gamma}) J_T K_T(\boldsymbol{\gamma}, \boldsymbol{\gamma})$, *which is correctly defined and is invertible.*

*Proof.* First, by the definition of the matrix $K_T(\boldsymbol{\gamma}, \boldsymbol{\gamma})$ and the vector $\mathcal{A}$, we have that the payment vector $\mathbf{r}(\boldsymbol{\gamma}, \boldsymbol{\gamma}, \mathcal{A}) = K_T(\boldsymbol{\gamma}, \boldsymbol{\gamma})\mathcal{A}$. Second, let us denote the intersection point of the lines $S_{\mathbf{a}^j}$ and $S_{\mathbf{a}^{j-1}}$ by $v_j$ for $j = 1, \ldots, k$ and combine them in the vector $\mathbf{v} = (v_1, \ldots, v_k)$. From the identities

$$\boldsymbol{\gamma} \cdot \mathbf{a}^j v_j - r_j(\boldsymbol{\gamma}, \boldsymbol{\gamma}, \mathcal{A}) = S_{\mathbf{a}^j}(v_j) = S_{\mathbf{a}^{j-1}}(v_j) = \boldsymbol{\gamma} \cdot \mathbf{a}^{j-1} v_j - r_j(\boldsymbol{\gamma}, \boldsymbol{\gamma}, \mathcal{A}), \qquad j = 1, \ldots, k,$$

by simple arithmetic calculations, one can show that these intersection points can be expressed via the payment vector in the following matrix form: $\mathbf{v} = Z_T(\boldsymbol{\gamma}) J_T \mathbf{r}(\boldsymbol{\gamma}, \boldsymbol{\gamma}, \mathcal{A})$. Combining with the previous finding, we have that $\mathbf{v} = Z_T(\boldsymbol{\gamma}) J_T K_T(\boldsymbol{\gamma}, \boldsymbol{\gamma})\mathcal{A}$. So, we obtain in this way the linear map $\mathbf{w}_{\boldsymbol{\gamma}}(\mathcal{A}) := W_T(\boldsymbol{\gamma})\mathcal{A} : \mathfrak{A}_T \to \mathbb{R}^k$ that depends on $\boldsymbol{\gamma}$.

Now we prove that $\mathbf{w}_{\boldsymbol{\gamma}}(\mathcal{A}) \in \Delta^{k(T)} \Leftrightarrow \mathcal{A} \in \tilde{\mathfrak{A}}_T(\boldsymbol{\gamma})$. We make via two following inductions.

- Let $\mathcal{A} \in \tilde{\mathfrak{A}}_T(\boldsymbol{\gamma})$ and $\mathbf{v} = \mathbf{w}_{\boldsymbol{\gamma}}(\mathcal{A})$. Then, for $j = 1, \ldots, k$, $v_j \geq 0$ (*the basis of the induction*). Indeed, assume that this condition is violated for some $j$, then $S_{\mathbf{a}^{j-1}}(v) < S_{\mathbf{a}^j}(v) \; \forall v > v_j$, but $v_j < 0$, and, thus $\mathbf{a}^{j-1}$ is not active, which is a contradiction. So, let us set $v_0 := 0$ (for the notation simplicity); assume, for $s \geq 0$, $0 \leq v_1 \leq \cdots \leq v_s$ and $v_s \leq v_{s+1}, \ldots, v_k$; and prove that $v_{s+1} \leq v_{s+2}, \ldots, v_k$ (*the inductive step*).

  Assume the contrary: for some $j > s + 1$ we have $v_s \leq v_j < v_{s+1}$. Then $S_{\mathbf{a}^s}(v_j) > S_{\mathbf{a}^{s+1}}(v_j)$ since $S_{\mathbf{a}^s}(v_{s+1}) = S_{\mathbf{a}^{s+1}}(v_{s+1})$ and the slope of $S_{\mathbf{a}^s}$ is less than that of $S_{\mathbf{a}^{s+1}}$. If $S_{\mathbf{a}^j}(v_j) \geq S_{\mathbf{a}^s}(v_j)$, we have $S_{\mathbf{a}^{s+1}}(v) < S_{\mathbf{a}^{j-1}}(v)$ for $v > v_j$ and $S_{\mathbf{a}^{s+1}}(v) < S_{\mathbf{a}^s}(v)$ for $v \leq v_j$, which means that $\mathbf{a}^{s+1}$ is not active. Otherwise, we have $S_{\mathbf{a}^{j-1}}(v) < S_{\mathbf{a}^s}(v)$ for $v \leq v_j$ and $S_{\mathbf{a}^{j-1}}(v) < S_{\mathbf{a}^j}(v)$ for $v > v_j$, which means that $\mathbf{a}^{j-1}$ is not active. Both cases infer contradiction, thus, the induction holds.

- Conversely, let $\mathbf{v} = \mathbf{w}_{\boldsymbol{\gamma}}(\mathcal{A}) \in \Delta^{k(T)}$. Then $S_{\mathbf{a}^j}(0) \leq S_{\mathbf{a}_{j-1}}(0)$ for all $j > 0$ (and, thus, $S_{\mathbf{a}^0}$ is active). Indeed, assume that this condition is violated for some $j$, then $v_j < 0$, contradiction (*the basis of the induction*). So, let us set $v_0 := 0$ (for the notation simplicity); assume, for $s \geq 0$, that

$$S_{\mathbf{a}^j}, j \leq s, \text{ are active}, \quad S_{\mathbf{a}^j}(v_s) \leq S_{\mathbf{a}^{j-1}}(v_s) \text{ for } j > s \quad \text{and} \quad S_{\mathbf{a}^j}(v_s) \leq S_{\mathbf{a}^s}(v_s) \text{ for } j < s;$$

and prove that

$$S_{\mathbf{a}^{s+1}} \text{is active}, S_{\mathbf{a}^j}(v_{s+1}) \leq S_{\mathbf{a}^{j-1}}(v_{s+1}) \text{for} j > s+1 \quad \text{and} \quad S_{\mathbf{a}^j}(v_{s+1}) \leq S_{\mathbf{a}^{s+1}}(v_{s+1}) \text{for} j < s+1;$$

i.e., (*the inductive step*).

The second condition is due to $v_j \geq v_{s+1}$ for $j > s + 1$. The third condition for $j = s$ follows from the definition of $v_{s+1}$ and the same for $j < s$ is due to the fact that $S_{\mathbf{a}^j}(v_{s+1}) \leq S_{\mathbf{a}^s}(v_{s+1})(= S_{\mathbf{a}_{s+1}}(v_{s+1}))$, since the slope of the function $S_{\mathbf{a}^j}$ is less than the slope of the function $S_{\mathbf{a}^s}$ and $S_{\mathbf{a}^j}(v_s) \leq S_{\mathbf{a}^s}(v_s)$. The second condition together with the third conditions gives the activeness of $S_{\mathbf{a}^{s+1}}$. Thus, the induction holds.

The matrices $Z_T$, $J_T$, and $K_T$ are invertible[5], thus, both the matrix $W_T$ and the map $\mathbf{w}_{\boldsymbol{\gamma}} : \mathfrak{A}_T \to \mathbb{R}^k$ are invertible as well. Hence, $\tilde{\mathfrak{A}}_T(\boldsymbol{\gamma})$ is linearly mapped onto $\Delta^{k(T)}$ by $\mathbf{w}_{\boldsymbol{\gamma}}$. $\qquad\square$

### A.2.3 Proof of Proposition 2

*Proof of Proposition 2.* Let us take the transformation $\mathbf{w}_{\boldsymbol{\gamma}^B}$ defined by $\mathbf{w}_{\boldsymbol{\gamma}^B}(\mathcal{A}) := W_T(\boldsymbol{\gamma}^B)\mathcal{A}$ (as in Lemma A.1 in Appendix A.2.2) and $\mathbf{v} = \mathbf{w}_{\boldsymbol{\gamma}^B}(\mathcal{A})$. Recall that, in this case, the $j$-th component of $\mathbf{v}$ is the intersection point of the straight-line functions $S_{\mathbf{a}^j}$ and $S_{\mathbf{a}^{j-1}}$. It is evident that the strategic buyer chooses the strategy $\mathbf{a}^j$, when his valuation $v$ is in the segment $[v_j; v_{j+1})$ for $j \geq 0$ (to be formally correct, we set $v_0 := 0, v_{k+1} := +\infty$). Thus, the expected strategic revenue equals to

$$\mathbb{E}\left[\mathrm{SRev}_{\boldsymbol{\gamma}^S, \boldsymbol{\gamma}^B}(\mathcal{A}, V)\right] = \sum_{j=1}^{k}(F_D(v_{j+1}) - F_D(v_j))(\boldsymbol{\gamma}^S \cdot \mathbf{p}(\mathbf{a}^j, \mathcal{A})) = \sum_{j=1}^{k}(F_D(v_{j+1}) - F_D(v_j))r_j(\boldsymbol{\gamma}^S, \boldsymbol{\gamma}^B, \mathcal{A}),$$

see the definitions of $\mathbf{p}$ and $\mathbf{r}$ before the proof of Lemma A.1. (Appendix A.2.2). Let us denote by $dF(\mathbf{v})$ the $k$-dimensional vector with $F_D(v_{j+1}) - F_D(v_j)$ in the $j$-th component, then, using the identity $dF(\mathbf{v}) = J_T^{\mathsf{T}}(1 - F_D(\mathbf{v}))$, we have

$$\mathbb{E}\left[\mathrm{SRev}_{\boldsymbol{\gamma}^S, \boldsymbol{\gamma}^B}(\mathcal{A}, V)\right] = dF(v)^{\mathsf{T}}\mathbf{r}(\boldsymbol{\gamma}^S, \boldsymbol{\gamma}^B, \mathcal{A}) = (1 - F_D(v))^{\mathsf{T}}J_T\mathbf{r}(\boldsymbol{\gamma}^S, \boldsymbol{\gamma}^B, \mathcal{A}).$$

From the definition of the matrix $K_T$, one can obtain $\mathbf{r}(\boldsymbol{\gamma}^S, \boldsymbol{\gamma}^B, \mathcal{A}) = K_T(\boldsymbol{\gamma}^B, \boldsymbol{\gamma}^S)\mathcal{A}$ (as in the proof of Lemma A.1. (Appendix A.2.2)). Finally, we have $\mathcal{A} = W_T(\boldsymbol{\gamma}^B)^{-1}\mathbf{v} = K_T(\boldsymbol{\gamma}^B, \boldsymbol{\gamma}^B)^{-1}J_T^{-1}Z_T(\boldsymbol{\gamma}^B)^{-1}\mathbf{v}$ due to $\mathbf{v} = W_T(\boldsymbol{\gamma}^B)\mathcal{A}$ and invertibility of $\mathbf{w}_{\boldsymbol{\gamma}^B}$.

So, let us combine all together:

$$\mathbb{E}\left[\mathrm{SRev}_{\boldsymbol{\gamma}^S, \boldsymbol{\gamma}^B}(\mathcal{A}, V)\right] = (1 - F_D(\mathbf{v}))^{\mathsf{T}}J_T \cdot K_T(\boldsymbol{\gamma}^B, \boldsymbol{\gamma}^S)K_T(\boldsymbol{\gamma}^B, \boldsymbol{\gamma}^B)^{-1}J_T^{-1}Z_T(\boldsymbol{\gamma}^B)^{-1}\mathbf{v},$$

where the matrix product between $(1 - F_D(\mathbf{v}))^{\mathsf{T}}$ and $\mathbf{v}$ is exactly the matrix $\Xi_T(\boldsymbol{\gamma}^S, \boldsymbol{\gamma}^B)$. $\qquad\square$

### A.2.4 The function $H_D$ as a special case of the functional $L_{D,\gamma^B,\gamma^B}$

Let us consider the case of equal discounts, $\gamma^S = \gamma^B$, then $K_T(\gamma^B, \gamma^S) = K_T(\gamma^B, \gamma^B)$ and the matrix $\Xi_T(\gamma^S, \gamma^B) = J_T \cdot K_T(\gamma^B, \gamma^S) K_T(\gamma^B, \gamma^B)^{-1} J_T^{-1} Z_T(\gamma^B)^{-1}$ becomes equal just to the diagonal matrix $Z_T(\gamma^B)^{-1} = \operatorname{diag}(\alpha_1, \ldots, \alpha_k)$, $\alpha_j = \gamma^B \cdot \mathbf{a}^j - \gamma^B \cdot \mathbf{a}^{j-1}$. Hence,

$$L_{D,\gamma^B,\gamma^B}(\mathbf{v}) = (1 - F_D(\mathbf{v}))^\mathsf{T} Z_T(\gamma^B)^{-1}\mathbf{v} = \sum_{j=1}^k (1 - F_D(v_j))\alpha_j v_j = \sum_{j=1}^k H_D(v_j)\alpha_j.$$

Since $\alpha_j > 0$ (due to the dependence of the order of $\{\mathbf{a}^j\}_j$ on $\gamma^B$) and $H_D(v) \le H_D(p_D^*)$, $\forall v$, (see Sec. 3) we infer that this sum above is maximal when $v_1 = \ldots = v_k = p_D^*$. Thus, in the case of equal discounts, the optimization of the functional $L_{D,\gamma^B,\gamma^B}$ reduces to the maximization of the function $H_D$ used to find Myerson's price $p_D^*$. This is expected and *additionally highlights the strong similarity of our optimization functional for the dynamic pricing to the one for the static pricing.*

## A.3 Missed Proofs from Section 5

### A.3.1 Proof of Proposition 3

*Proof of Proposition 3.* The left inequality is trivial, since $\mathfrak{A}_\infty^\tau \subset \mathfrak{A}_\infty$. The second obvious observation is that $\mathrm{SRev}_{\gamma^S,\gamma^B}(\mathcal{A}, v)$ for $\mathcal{A} \in \mathfrak{A}_\infty^\tau$ is equal to $\mathrm{SRev}_{\gamma^{S,\tau},\gamma^{B,\tau}}(\mathcal{A}^\tau, v)$, where $\gamma^{S,\tau} = (\gamma_1^S, \ldots, \gamma_{\tau-1}^S, \sum_{t=\tau}^\infty \gamma_t^S)$, $\gamma^{B,\tau} = (\gamma_1^B, \ldots, \gamma_{\tau-1}^B, \sum_{t=\tau}^\infty \gamma_t^B)$ and $\mathcal{A}^\tau \in \mathfrak{A}_\tau$ is a restriction of $\mathcal{A}$ on $\{\mathfrak{n} \mid |\mathfrak{n}| \le \tau - 1\}$. Thus,

$$\max_{\mathcal{A} \in \mathfrak{A}_\infty^\tau} \mathbb{E}\left[\mathrm{SRev}_{\gamma^S,\gamma^B}(\mathcal{A}, V)\right] = \max_{\mathcal{A} \in \mathfrak{A}_\tau} \mathbb{E}\left[\mathrm{SRev}_{\gamma^{S,\tau},\gamma^{B,\tau}}(\mathcal{A}, V)\right].$$

The following step of the proof is formulated as a lemma:

**Lemma A.2.** *Let $\gamma^S, \gamma^{S,1}, \gamma^{S,2}$ be discounts such that $\gamma^S = \gamma^{S,1} + \gamma^{S,2}$.* [6] *In this case*

$$\max_{\mathcal{A} \in \mathfrak{A}} \mathbb{E}\left[\mathrm{SRev}_{\gamma^S,\gamma^B}(\mathcal{A}, V)\right] \le \max_{\mathcal{A} \in \mathfrak{A}} \mathbb{E}\left[\mathrm{SRev}_{\gamma^{S,1},\gamma^B}(\mathcal{A}, V)\right] + \max_{\mathcal{A} \in \mathfrak{A}} \mathbb{E}\left[\mathrm{SRev}_{\gamma^{S,2},\gamma^B}(\mathcal{A}, V)\right]$$

We omit the proof, since it is trivial. Apply Lemma A.2 to the sellers discount divided into two parts as follows: $\gamma^S = \gamma^S \cdot I_{\{t > \tau\}} + \gamma^S \cdot I_{\{t \le \tau\}}$:

$$\max_{\mathcal{A} \in \mathfrak{A}} \mathbb{E}\left[\mathrm{SRev}_{\gamma^S,\gamma^B}(\mathcal{A}, V)\right] \le \max_{\mathcal{A} \in \mathfrak{A}} \mathbb{E}\left[\mathrm{SRev}_{\gamma^S \cdot I_{\{t \le \tau\}},\gamma^B}(\mathcal{A}, V)\right] + \max_{\mathcal{A} \in \mathfrak{A}} \mathbb{E}\left[\mathrm{SRev}_{\gamma^S \cdot I_{\{t > \tau\}},\gamma^B}(\mathcal{A}, V)\right]$$

The left term of the right-hand side of the inequality is not greater than $\max_{\mathcal{A} \in \mathfrak{A}_\tau} \mathbb{E}\left[\mathrm{SRev}_{\gamma^{S,\tau},\gamma^{B,\tau}}(\mathcal{A}, V)\right]$, since $\gamma^{S,\tau} \ge \gamma^S \cdot I_{\{t \le \tau\}}$ for $t \le \tau$ and for $t > \tau$ the discount $\gamma^S \cdot I_{\{t \le \tau\}}$ is zero. The right term is not greater than $\Gamma_\tau^S \mathbb{E}[V]$. This fact can be proved in following several steps:

1. Following identitiy can be verified by the direct application of $\nu(\gamma)$ definition:

$$\gamma_{\tau+i}^S = \gamma_{\tau+i}^B \cdot \frac{\nu(\gamma^S)_{\tau+i-1}}{\nu(\gamma^B)_{\tau+i-1}} \cdot \ldots \cdot \frac{\nu(\gamma^S)_{\tau+1}}{\nu(\gamma^B)_{\tau+1}} \cdot \frac{\gamma_\tau^S}{\gamma_\tau^B}$$

2. Define $c_i := \frac{\nu(\gamma^S)_{\tau+i-1}}{\nu(\gamma^B)_{\tau+i-1}} \cdot \ldots \cdot \frac{\nu(\gamma^S)_{\tau+1}}{\nu(\gamma^B)_{\tau+1}} \cdot \frac{\gamma_\tau^S}{\gamma_\tau^B}$ for $i \ge 1$, thus, $c_i$ are increasing and

$$\gamma_{\tau+i}^S = c_i \cdot \gamma_{\tau+i}^B$$

3. In this case

$$\boldsymbol{\gamma}^{\mathsf{S}} \cdot \mathbb{I}_{\{t>\tau\}} = c_1 \boldsymbol{\gamma}^{\mathsf{B}} \cdot \mathbb{I}_{\{t>\tau\}} + (c_2 - c_1)\boldsymbol{\gamma}^{\mathsf{B}} \cdot \mathbb{I}_{\{t>\tau+1\}} + (c_3 - c_2)\boldsymbol{\gamma}^{\mathsf{B}} \cdot \mathbb{I}_{\{t>\tau+2\}}$$

4. Consider a case when the discount of the seller is $c \cdot \boldsymbol{\gamma}^{\mathsf{B}} \cdot \mathbb{I}_{\{t>\tau+i\}}$ for some $c > 0$. Let $\mathcal{A} \in \mathfrak{A}$ and $\mathbf{a} = \{a_t\}_{t=1}^{\infty}$ be some optimal strategy for a valuation $v > 0$. Then

$$S(v) = \sum_{t=1}^{\infty} a_t \gamma_t^{\mathsf{B}}(v - \mathcal{A}(\mathbf{a}_{1:t-1})) \geq \sum_{t=1}^{\tau+i} a_t \gamma_t^{\mathsf{B}}(v - \mathcal{A}(\mathbf{a}_{1:t-1})) \Rightarrow \sum_{t=\tau+i+1}^{\infty} a_t \gamma_t^{\mathsf{B}} v \geq \sum_{t=\tau+i+1}^{\infty} a_t \gamma_t^{\mathsf{B}} \mathcal{A}(\mathbf{a}_{1:t-1})$$

But the right part of the last inequality is $\frac{1}{c}\mathrm{SRev}_{c\boldsymbol{\gamma}^{\mathsf{B}} \cdot \mathbb{I}_{\{t>\tau+i\}}, \boldsymbol{\gamma}^{\mathsf{B}}}(\mathcal{A}, v)$, and, thus,

$$c\Gamma_{\tau+i}^{\mathsf{B}} v \geq \mathrm{SRev}_{c\boldsymbol{\gamma}^{\mathsf{B}} \cdot \mathbb{I}_{\{t>\tau+i\}}, \boldsymbol{\gamma}^{\mathsf{B}}}(\mathcal{A}, v) \Rightarrow c\Gamma_{\tau+i}^{\mathsf{B}} \mathbb{E}\left[V\right] \geq \max_{\mathcal{A} \in \mathfrak{A}} \mathbb{E}\left[\mathrm{SRev}_{c\boldsymbol{\gamma}^{\mathsf{B}} \cdot \mathbb{I}_{\{t>\tau+i\}}, \boldsymbol{\gamma}^{\mathsf{B}}}(\mathcal{A}, V)\right]$$

5. Finally, apply Lemma A.2 and the identity from our third step and get (for the notation simplicity $c_0 := 0$):

$$\max_{\mathcal{A} \in \mathfrak{A}} \mathbb{E}\left[\mathrm{SRev}_{\boldsymbol{\gamma}^{\mathsf{S}} \cdot I_{\{t>\tau\}}, \boldsymbol{\gamma}^{\mathsf{B}}}(\mathcal{A}, V)\right] \leq \sum_{i=1}^{\infty} (c_i - c_{i-1})\Gamma_{\tau+i-1}^{\mathsf{B}} \mathbb{E}\left[V\right] = \Gamma_{\tau}^{\mathsf{S}} \mathbb{E}\left[V\right] \qquad \text{Q.E.D.}$$

$\square$

## A.4    Existence of an optimal strategy $\mathbf{a} \in \mathfrak{S}_T$

Assume we are given an algorithm $\mathcal{A} \in \mathfrak{A}_T$, a correct discount sequence $\boldsymbol{\gamma} = \{\gamma_t\}_{t=1}^{T}$ and a private valuation $v \in [0; +\infty)$ (they are fixed). In this case, for the function $F : \mathfrak{S}_T \to \mathbb{R} \cup \{-\infty\}$, $F(\mathbf{a}) = S_{\mathbf{a}}(v)$ the following proposition holds.

**Proposition A.3.** *There exists a strategy $\mathbf{a}^* \in \mathfrak{S}_T$ such that $\forall \mathbf{a} \in \mathfrak{S}_T : F(\mathbf{a}^*) \geq F(\mathbf{a})$.*

*Proof of Proposition A.3.* The proof of the case $T < \infty$ is trivial: we just need to choose the maximum of a finite set of numbers. Thus, let us consider $T = \infty$.

Denote $M = \sup_{\mathbf{a} \in \mathfrak{S}_{\infty}} S_{\mathbf{a}}(v)$ and $\mathfrak{S}^0 = \mathfrak{S}_{\infty}$. Let $k \geq 0$ be a non-negative integer, then assume that $a_1^*, \ldots, a_k^* \in \{0, 1\}$ and $\mathfrak{S}^k = \{\mathbf{a} = \{a_t\}_{t=1}^{\infty} \in \mathfrak{S} | a_1 \ldots a_k = a_1^* \ldots a_k^*\}$ such that $\sup_{\mathbf{a} \in \mathfrak{S}^k} S_{\mathbf{a}}(v) = M$ defined (if such conditions holds we call the tuple $(a_1^*, \ldots, a_k^*, \mathfrak{S}^k)$ *correct*). We define such $a_{k+1}^*$ that the tuple $(a_1^*, \ldots, a_k^*, a_{k+1}^*, \mathfrak{S}^{k+1})$ is correct.

$\{\mathbf{a} = \{a_t\}_{t=1}^{\infty} \in \mathfrak{S}_{\infty} | a_1 \ldots a_k a_{k+1} = a_1^* \ldots a_k^* 0\}$ and $\{\mathbf{a} = \{a_t\}_{t=1}^{\infty} \in \mathfrak{S}_{\infty} | a_1 \ldots a_k a_{k+1} = a_1^* \ldots a_k^* 1\}$ are denoted by $\mathfrak{S}^{0,k}$ and $\mathfrak{S}^{1,k}$ respectively, similarly $\sup_{\mathbf{a} \in \mathfrak{S}^{0,k}} S_{\mathbf{a}}(v)$ and $\sup_{\mathbf{a} \in \mathfrak{S}^{1,k}} S_{\mathbf{a}}(v)$ are denoted by $M^0$ and $M^1$. Since $\mathfrak{S}^k = \mathfrak{S}^{0,k} \cup \mathfrak{S}^{1,k}$, the following identity holds $M = \max\{M^0, M^1\}$. If $M = M^0$ we define $a_{k+1}^* = 0$ and $\mathfrak{S}^{k+1} = \mathfrak{S}^{0,k}$, otherwise $a_{k+1}^* = 1$ and $\mathfrak{S}^{k+1} = \mathfrak{S}^{1,k}$. Thus, by the definition of $a_{k+1}^*$ and $\mathfrak{S}^{k+1}$ the tuple $(a_1^*, \ldots, a_k^*, a_{k+1}^*, \mathfrak{S}^{k+1})$ is correct.

Taking into account that for $k = 0$ the tuple $(\mathfrak{S}^0)$ is correct, we obtain uniquely defined sequence $\mathbf{a}^* = \{a_t^*\}_{t=1}^{\infty}$ such that for all integer $k \geq 0$ the tuple $(a_1^*, \ldots, a_k^*, \mathfrak{S}^k)$ for $\mathfrak{S}^k = \{\mathbf{a} = \{a_t\}_{t=1}^{\infty} \in \mathfrak{S}_{\infty} | a_1 \ldots a_k = a_1^* \ldots a_k^*\}$ is correct.

Before finally proving Proposition 1 we prove an auxillary statement:

$$\forall \tau \in \mathbb{N} \quad \sum_{t \geq \tau} \gamma_t a_t^* (v - p_t) \geq 0,$$

where $\{p_t\}_{t=1}^{\infty}$ is the sequence of prices set by the algorithm $\mathcal{A}$ in response to $\mathbf{a}^*$. Indeed, for an arbitrary $\tau \in \mathbb{N}$ assume $\sum_{t \geq \tau} \gamma_t a_t^*(v - p_t) = -\delta < 0$. In this case, since the series $\sum_{t \geq \tau} \gamma_t$ and $\sum_{t \geq \tau} \gamma_t a_t^*(v - p_t)$ converge, there exists such integer $\tau_0 \geq \tau$ that $\sum_{t \geq \tau_0} \gamma_t v < \frac{\delta}{3}$ and $\sum_{t \geq \tau_0} \gamma_t a_t^*(v - p_t) > -\frac{\delta}{2}$, which impllies

$$\sum_{\tau_0 \geq t \geq \tau} \gamma_t a_t^*(v - p_t) = \sum_{t \geq \tau} \gamma_t a_t^*(v - p_t) - \sum_{t > \tau_0} \gamma_t a_t^*(v - p_t) < -\delta + \frac{\delta}{2} = -\frac{\delta}{2}$$

Thus, for an arbitrary strategy $\mathbf{a} \in \mathfrak{S}^{\tau_0}$ (denote prices corresponding to $\mathbf{a}$ by $q_t : \forall t \leq \tau_0 + 1 : q_t = p_t$) we gain for $\mathbf{b} = a_1^* \dots a_\tau^* 0^\infty$

$$S_{\mathbf{a}}(v) = \sum_{t \leq \tau} \gamma_t a_t^*(v - p_t) + \sum_{\tau < t \leq \tau_0} \gamma_t a_t^*(v - p_t) + \sum_{\tau_0 < t} \gamma_t a_t(v - q_t) < S_{\mathbf{b}}(v) - \frac{\delta}{2} + \frac{\delta}{3} < M - \frac{\delta}{6},$$

which implies $\sup_{\mathbf{a} \in \mathfrak{S}^{\tau_0 - 1}} S_{\mathbf{a}}(v) \leq M - \frac{\delta}{6} < M$. This contradicts to the correctness of the tuple $(a_1^*, \dots, a_{\tau_0 - 1}^*, \mathfrak{S}^{\tau_0 - 1})$.

Now assume $\forall \mathbf{a} \in \mathfrak{S}_\infty \ S_{\mathbf{a}}(v) < M$, hence, $S_{\mathbf{a}^*}(v) < M$ and there exists such strategy $\mathbf{a} \in \mathfrak{S}_\infty$ that $M > S_{\mathbf{a}}(v) > S_{\mathbf{a}^*}(v)$ by the definition of $M$. Define $\varepsilon = S_{\mathbf{a}}(v) - S_{\mathbf{a}^*}(v)$. Consider an integer $\tau_1 \geq 0$ such that $\sum_{t > \tau_1} \gamma_t v < \varepsilon$ ($\tau_1$ exists, since the series $\sum_{t=1}^{\infty} \gamma_t$ converges). By the definition of $\mathbf{a}^*$ and $\mathfrak{S}^{\tau_1}$ there exists such strategy $\mathbf{b} = \{b_t\}_{t=1}^{\infty} \in \mathfrak{S}^{\tau_1}$ that $S_{\mathbf{b}}(v) > S_{\mathbf{a}}(v)$ (and $b_1 \dots b_{\tau_1} = a_1^* \dots a_{\tau_1}^*$). Hence, denoting the price sequence set by $\mathcal{A}$ in response to $\mathbf{b}$ by $\{p_t^1\}_{t=1}^{\infty}$ ($p_1^1 \dots p_{\tau_1+1}^1 = p_1 \dots p_{\tau_1+1}$) and using that $\sum_{t > \tau_1} \gamma_t a_t^*(v - p_t) \geq 0$ we gain

$$\varepsilon < S_{\mathbf{b}}(v) - S_{\mathbf{a}^*}(v) = \sum_{t > \tau_1} \gamma_t b_t(v - p_t^1) - \sum_{t > \tau_1} \gamma_t a_t^*(v - p_t) < \varepsilon - 0 = \varepsilon,$$

which is the contradiction. Thus, the desired result $\exists \mathbf{a} \in \mathfrak{S} \ S_{\mathbf{a}}(v) = M$ is obtained. $\square$

An optimal strategy can be not unique here though, and, therefore, the strategic revenue (see Sec.2) is not always defined. This is not a problem for us due to Corollary B.1: the optimal strategy is unique for almost all $v$, and since we have continuous distribution, the ESR (expected strategic revenue) is not affected by the indefiniteness of SRev on the set of zero measure.

## A.5  Existence of a global maximum point of $H_D(v)$

Assume the non-negative random variable $V \sim D$ has a finite expectation, and the distribution function $G(v) = \mathbb{P}_{V \sim D}[v < V]$ is continuous.

**Proposition A.4.** *In this case, the function $H_D(v) = G(v) \cdot v$ has a global maxima point.*

*Proof of Proposition A.4.* We denote the probability measure function by $\mu_V : \mathfrak{B}(\mathbb{R}) \to [0; 1]$, $\mu_V(A) = \mathbb{P}[V \in A]$, since we already use $\mathbb{P}$ in traditional manner (e.g. $\mathbb{P}[V \geq 0]$). Here $\mathfrak{B}(\mathbb{R})$ is the Borel Algebra for $\mathbb{R}$ with the standard topology set. In this terms, the function $H_D$ can be expressed as follows

$$H_D(\bar{v}) = G(\bar{v}) \cdot \bar{v} = \mathbb{P}[V > \bar{v}] \cdot \bar{v} = \bar{v} \cdot \int_{(\bar{v};+\infty)} 1 d\mu_V,$$

which can be upper bound by $\int_{(\bar{v};+\infty)} v d\mu_V$, where $\int_A f(v) d\mu_V$ denotes the Lebegue's integral of a function $f$ on a set $A$ w.r.t. the probability measure $\mu_V$. Due to the absolute continuity of the

Lebesgue integral (see Appendix H.1), the fact that $\mu_V((\bar{v}; +\infty)) \xrightarrow[\bar{v} \to \infty]{} 0$ holds, and, since $V$ has a finite expectation, we obtain that $\int_{(\bar{v}; +\infty)} v d\mu_V \xrightarrow[\bar{v} \to \infty]{} 0$ and, thus,

$$H_D(\bar{v}) \leq \int_{(\bar{v}; +\infty)} v d\mu_V \xrightarrow[\bar{v} \to \infty]{} 0 \Rightarrow H_D(\bar{v}) \xrightarrow[\bar{v} \to \infty]{} 0.$$

Hence, for an arbitrary picked point $v_0 > 0$ such that $H_D(v_0) > 0$ there exists such $v_1 > v_0$ that $\forall v > v_1 \ H_D(v) < H_D(v_0)$, thus, if $H_D$ has a maxima point $v^*$ in the segment $[0; v_1]$, then $v^*$ is the global maxima point, but $H_D$ is a continuous function, hence, it has a maxima point in any segment, thus, the proposition is proved.

$\square$

# B  Alternative proof for the case $\gamma^{\mathsf{S}} = \gamma^{\mathsf{B}}$

Let $\gamma := \gamma^{\mathsf{S}} = \gamma^{\mathsf{B}}$ all throughout this section. The proof holds for any discounts, not only geometric ones. The game horizon $T$ may be both finite and infinite. For simplicity, we use the following short form notations: $\mathrm{SRev}_{\gamma} := \mathrm{SRev}_{\gamma, \gamma}$. First of all, we summarize some useful properties of surplus and revenue as functions of the valuation $v$.

**Remark B.1.** Let a pricing algorithm $\mathcal{A} \in \mathfrak{A}_T$ and the discount sequence $\gamma$ be given. Denote $S_{\mathbf{a}}(v) := \mathrm{Sur}_{\gamma}(\mathcal{A}, v, \mathbf{a})$ and $S(v) := \mathrm{Sur}_{\gamma}(\mathcal{A}, v, \mathbf{a}^{Opt}(\mathcal{A}, v, \gamma))$. These functions satisfy following properties:

1. for each strategy $\mathbf{a} \in \mathfrak{S}_T$, the surplus $S_{\mathbf{a}}$ w.r.t. this strategy is a linear function of $v$ of the form $S_{\mathbf{a}}(v) = q_{\mathbf{a}} v - r_{\mathbf{a}}$, where $q_{\mathbf{a}} := \sum_{t=1}^{T} \gamma_t a_t (= \gamma \cdot \mathbf{a})^7$ is the *discounted quantity* of purchased goods and $r_{\mathbf{a}}$ is the discounted revenue of the seller (i.e., $r_{\mathbf{a}} = \mathrm{Rev}_{\gamma}(\mathcal{A}, \mathbf{a})$);

2. the strategic (optimal) surplus $S$ is convex as a function of $v$, because it is the maximum of a set of linear functions: $S(v) = \max_{\mathbf{a} \in \mathfrak{S}_T} S_{\mathbf{a}}(v)$ (by definition);

3. the strategic surplus $S(v)$ is non-negative for any $v \geq 0$ since, for the strategy $\mathbf{a} = \mathbf{0}^T$, we have $S_{\mathbf{a}}(v) = 0$, which implies in turn that $S(v) \geq S_{\mathbf{a}}(v) = 0$, $\forall v \geq 0$;

4. the derivative $S'(v)$ exists for almost all $v \in [0; +\infty)$ (i.e., it does not exist on a set of Lebesgue measure zero), because $S(v)$ is convex and is thus absolutely continuous.

**Lemma B.1.** *For any pricing algorithm $\mathcal{A} \in \mathfrak{A}_T$, the strategic revenue $R(v) := \mathrm{SRev}_{\gamma}(\mathcal{A}, v)$ is increasing on the valuation domain $[0; +\infty)$, it starts from zero (i.e., $R(0) = 0$), and the random variable $R(V)$ has thus finite non-negative expectation (i.e., $0 \leq \mathbb{E}[R(V)] < +\infty$).*

*Proof.* Proof of that $R(v)$ is increasing. For any two valuations $v_1$ and $v_2 \in [0; +\infty)$ s.t. $v_1 < v_2$, and two corresponding optimal strategies $\mathbf{a}^1$ and $\mathbf{a}^2 \in \mathfrak{S}_T$, i.e., such that $S(v_j) = S_{\mathbf{a}^j}(v_j)$, $j = 1, 2$, (using the notations from Remark B.1), we have

$$S_{\mathbf{a}^1}(v_1) \geq S_{\mathbf{a}^2}(v_1) \quad \text{and} \quad S_{\mathbf{a}^2}(v_2) \geq S_{\mathbf{a}^1}(v_2).$$

Therefore, since $S_{\mathbf{a}^j}, j = 1, 2$, are linear, they either coincide (then $r_{\mathbf{a}^1} = r_{\mathbf{a}^2}$), or have an intersection point $w$ in $[v_1, v_2] \subset [0; +\infty)$. In the latter case, one gets $S_{\mathbf{a}^1}(v) \geq S_{\mathbf{a}^2}(v) \ \forall v \in [0, w]$, which implies $-r_{\mathbf{a}^1} \geq -r_{\mathbf{a}^2}$ when $v = 0$. Hence, we obtain $R(v_2) = r_{\mathbf{a}^2} \geq r_{\mathbf{a}^1} = R(v_1)$ for any $v_2 > v_1 \geq 0$.

Proof of $R(0) = 0$.

Now we also note that $\forall v \geq 0 \;\; R(v) \geq 0$, since $R(v) = \sum_{t=1}^{T} \gamma_t a_t p_t$ (as a sum of non-negative terms). By the definition $S(0) = -R(0)$ and, thus, $R(0) = -S(0) \leq 0$. Therefore, $R(0) = 0$.

Proof of $0 \leq \mathbb{E}\left[R(V)\right] < +\infty$.

Now consider $v \geq 0$ and let $\mathbf{a} = \{a_t\}_{t=1}^{T}$ be the optimal strategy for $v$. Hence,

$$0 \leq S(v) = \sum_{t=1}^{T} \gamma_t a_t (v - p_t) \leq \Gamma v - R(v) \Rightarrow R(v) \leq \Gamma v$$

Thus,

$$\forall v \geq 0: \; 0 \leq R(v) \leq \Gamma v \Rightarrow 0 \leq \mathbb{E}\left[R(V)\right] \leq \Gamma \cdot \mathbb{E}\left[V\right]$$

$\square$

Similarly to the optimal surplus function $S(\cdot)$ and the strategic revenue one $R(\cdot)$, we introduce *the strategic purchased quantity* $Q(\cdot)$ as a map from the valuation domain, i.e., $Q(v) := \sum_{t=1}^{T} \gamma_t a_t^O(v)$, where $\{a_t^O(v)\}_{t=1}^{\infty} = \mathbf{a}^{Opt}(\mathcal{A}, v, \boldsymbol{\gamma})$. Note that $S(v) = Q(v)v - R(v)$, for each $v \in [0, +\infty)$.

**Lemma B.2.** *Assume that, for a given $v \geq 0$, the derivative $S'(v)$ exists. Then, $Q(v)$ is uniquely defined and equals to $S'(v)$ for any optimal strategy $\mathbf{a}$ of the buyer that holds the valuation $v$.*

*Proof.* Consider an arbitrary optimal strategy $\mathbf{a}$ for the valuation $v$, i.e., s.t. $S(v) = S_{\mathbf{a}}(v)$. Since $S_{\mathbf{a}}(w) = q_{\mathbf{a}} w - r_{\mathbf{a}}$ for any $w \geq 0$ and $S_{\mathbf{a}}(v) = S(v)$, we can write $S_{\mathbf{a}}(v + \delta) = S(v) + q_{\mathbf{a}} \delta$ and by the definition of the derivative $S(v + \delta) = S(v) + S'(v)\delta + o_{\delta \to 0}(\delta)$, thus $S_{\mathbf{a}}(v + \delta) - S(v + \delta) = (q_{\mathbf{a}} - S'(v))\delta + o_{\delta \to 0}(\delta)$, which should be not greater than zero for all possible $\delta$ since $S$ is convex. Hence we get $q_{\mathbf{a}} = S'$, because otherwise $(q_{\mathbf{a}} - S'(v))\delta + o_{\delta \to 0}(\delta)$ will take both positive and negative values in a neighborhood of 0. Finally, remind that $Q(v) = q_{\mathbf{a}}$ since $\mathbf{a}$ is optimal for $v$. $\square$

Lemma B.2 together with the identity $\mathrm{SRev}_{\boldsymbol{\gamma}}(\mathcal{A}, v) = R(v) = Q(v)v - S(v)$ gives us:

**Corollary B.1.** *For almost all $v \in [0; +\infty)$, the strategic revenue $\mathrm{SRev}_{\boldsymbol{\gamma}}(\mathcal{A}, v)$ is uniquely defined for any optimal strategy $\mathbf{a}$ of the buyer that holds the valuation $v$.*

**Remark B.2.** Function $Q(v)$ is defined almost everywhere and non-decreasing on its domain, since $Q'(v) = S''(v)$, which also defined almost everywhere and not less than 0, since $S$ is convex on its domain[8]. Also by the definition $Q(v) \leq \Gamma$ and, thus, $Q(+\infty)$ is finite.

## B.1 Optimality of the constant algorithm with the Myerson price

We use following notations for the distribution functions: $F(v) := \mathbb{P}[V \leq v]$ and $G(v) := 1 - F(v)$.

**Lemma B.3.** *For the mappings $S(v)$, $R(v)$, and $Q(v)$ the following identity holds:*

$$\mathbb{E}\left[R(V)\right] = \int_{[0;+\infty)} G(v)Q(v)dv + \int_{[0;+\infty)} G(v)v\,dQ(v) - \int_{[0;+\infty)} G(v)dS(v).$$

[8]Note that this fact can be proved directly like in Lemma B.1.

*Proof.* First we note, that due to the absolute continuity of the Lebesgue integral (see Appendix H.1) and the fact that $R(V)$ has a finite expectation we can write $\mathbb{E}\left[R(V)I_{[0;\bar{v}]}(V)\right] \xrightarrow[\bar{v}\to+\infty]{} \mathbb{E}\left[R(V)\right]$. Rewrite $\mathbb{E}\left[R(V)I_{[0;\bar{v}]}(V)\right]$ using Lebesgue-Stieltjes integral w.r.t. the fact that $F(v) = 1 - G(v)$:

$$\mathbb{E}\left[R(V)I_{[0;\bar{v}]}(V)\right] = \int\limits_{[0;\bar{v}]} R(v)dF(v) = -\int\limits_{[0;\bar{v}]} R(v)dG(v) \tag{B.1}$$

For the latter integral we use the integration by parts formula (which holds since $G$ is continuous on it's domain and $R$ is non-decreasing on it's domain) and gain

$$-\int\limits_{[0;\bar{v}]} R(v)dG(v) = -G(v)R(v)\bigg|_0^{\bar{v}} + \int\limits_{[0;\bar{v}]} G(v)dR(v) = \tag{B.2}$$

$$-G(v)R(v)\bigg|_0^{\bar{v}} + \int\limits_{[0;\bar{v}]} G(v)d(Q(v)\cdot v) - \int\limits_{[0;\bar{v}]} G(v)dS(v) \tag{B.3}$$

Since the function $G$ is continuous on $[0;\bar{v}]$, the Riemann-Stieltjes integral $\int\limits_0^{\bar{v}} G(v)d(Q(v)\cdot v)$ is defined and equals to the corresponding Lebesgues-Stieltjes integral. For the Riemann-Stieltjes integral for our conditions following identity holds

$$\int\limits_0^{\bar{v}} G(v)d(Q(v)\cdot v) = \int\limits_0^{\bar{v}} G(v)Q(v)dv + \int\limits_0^{\bar{v}} G(v)\cdot vdQ(v). \tag{B.4}$$

Bringing together that $R(0) = 0$ (Lemma 1) and $\lim\limits_{\bar{v}\to+\infty} R(\bar{v})G(\bar{v}) = 0$ (because

$$R(\bar{v})G(\bar{v}) \leq \int\limits_{(\bar{v};+\infty)} R(v)dF(v) \xrightarrow[\bar{v}\to+\infty]{} 0,$$

which holds due to the absolute continuity of the Lebesgue integral), Eq. (B.1), Eq. (B.2) and Eq. (B.4) (and taking the limits) we get

$$\mathbb{E}\left[R(V)\right] = \lim\limits_{\bar{v}\to+\infty} \mathbb{E}\left[R(V)I_{[0;\bar{v}]}(V)\right] = \int\limits_0^{+\infty} G(v)Q(v)dv + \int\limits_0^{+\infty} G(v)vdQ(v) - \int\limits_{[0;+\infty)} G(v)dS(v).$$

Considering first two integrals in the latter expression as Lebesgue and Lebesgue-Stieltjes integrals respectively, we obtain the desired result

$\square$

Finally we prove the bound in Eq.(3) which we stated in Sec.3:

**Theorem B.1.** *Assume the valuation $V \sim D$. Then the following bound holds:*

$$\mathbb{E}\left[\text{SRev}_{\gamma,\gamma}(\mathcal{A}, V)\right] \leq \Gamma \cdot H_D(p_D^*) \qquad \forall \mathcal{A} \in \mathfrak{A}_T. \tag{B.5}$$

*Proof.* Consider an arbitrary algorithm $\mathcal{A} \in \mathfrak{A}$ and use the notations $S$, $R$, and $Q$ introduced above. From Lemma B.3, we have

$$\mathbb{E}\left[R(V)\right] = \int\limits_{[0;+\infty)} G(v)Q(v)dv + \int\limits_{[0;+\infty)} G(v)vdQ(v) - \int\limits_{[0;+\infty)} G(v)dS(v) = \int\limits_{[0;+\infty)} G(v)vdQ(v), \tag{B.6}$$

where the latter identity of Eq. (B.6) holds due to the facts that $S$ is absolutely continuous on its domain (see Remark B.1), thus, $\int_{[0;+\infty)} G(v)dS(v) = \int_{[0;+\infty)} G(v)S'(v)dv$, and that $S'(v) = Q(v)$ almost everywhere (see Lemma B.2). By definition, we have $H_D(v) = G(v)v, \ \forall v \geq 0$, and, hence, Eq. (B.6) implies that $\mathbb{E}\left[R(V)\right] = \int_{[0;+\infty)} H_D(v)dQ(v)$ can be upper bound by the expression

$$H_D(p_D^*) \cdot \int_{[0;+\infty)} 1 dQ(v) = H_D(p_D^*) \cdot (Q(+\infty) - Q(0)) \leq H_D(p_D^*) \cdot \Gamma, \tag{B.7}$$

where $H_D(v)$ is bounded by its maximum $H_D(p_D^*)$, the first identity is due to the fact that $Q$ is non-decreasing on $v$, and non-negative $Q(v)$ is bounded by $\Gamma$ for all $v \geq 0$ (see Remark B.2). Thus, the theorem is proved. □

# C    Numerical solutions for finite games and different distributions of $V$

## C.1    The dimension reduction in the case of finite game with horizon $T = 2$

Let $\boldsymbol{\gamma}^{\mathsf{B}} = \{\gamma_{\mathsf{B}}^{t-1}\}_{t=1}^2$ and $\boldsymbol{\gamma}^{\mathsf{S}} = \{\gamma_{\mathsf{S}}^{t-1}\}_{t=1}^2$ for $0 < \gamma_{\mathsf{B}} < \gamma_{\mathsf{S}} < 1$. Build the $\Xi_T(\boldsymbol{\gamma}^{\mathsf{B}}, \boldsymbol{\gamma}^{\mathsf{S}})$ matrix by the definition:

$$\Xi_T(\boldsymbol{\gamma}^{\mathsf{B}}, \boldsymbol{\gamma}^{\mathsf{S}}) = \begin{pmatrix} \gamma_{\mathsf{S}} & 0 & 0 \\ -(\gamma_{\mathsf{S}} - \gamma_{\mathsf{B}}) & 1 - \gamma_{\mathsf{B}} & 0 \\ 0 & 0 & \gamma_{\mathsf{S}} \end{pmatrix}$$

Now we prove that $L(v) = (1 - F(v))^{\mathsf{T}} \Xi_T(\boldsymbol{\gamma}^{\mathsf{B}}, \boldsymbol{\gamma}^{\mathsf{S}})v$ always has a maximum on the hyperplane $v_2 = v_3$.

Indeed, assume it's not and consider a maximum point with coordinates $v_1, v_2, v_3$ such that $v_2 < v_3$. Consider two possible cases: $(1 - F(v_3))v_3 > (1 - F(v_2))v_2$ and $(1 - F(v_3))v_3 \leq (1 - F(v_2))v_2$.

- $(1 - F(v_3))v_3 > (1 - F(v_2))v_2$ implies $L(v_1, v_2, v_3) < L(v_1, v_3, v_3)$, since

  $$L(v_1, v_3, v_3) - L(v_1, v_2, v_3) = -(\gamma_{\mathsf{S}} - \gamma_{\mathsf{B}})(F(v_2) - F(v_3))v_1 + (1 - \gamma_{\mathsf{B}})((1 - F(v_3))v_3 - (1 - F(v_2))v_2),$$

  where the left term is non-negative, since $v_3 > v_2$, and the right term is strictly positive by the assumption. $L(v_1, v_2, v_3) < L(v_1, v_3, v_3)$ contradicts to our assumption.

- $(1 - F(v_3))v_3 \leq (1 - F(v_2))v_2$ implies $L(v_1, v_2, v_3) \leq L(v_1, v_2, v_2)$, since

  $$L(v_1, v_2, v_2) - L(v_1, v_2, v_3) = \gamma_{\mathsf{S}}((1 - F(v_2))v_2 - (1 - F(v_3))v_3) \geq 0.$$

  This also contradicts to our assumption.

Both possible cases infer contradiction, thus, our assumption is wrong. Q.E.D.

## C.2    Experiments

In all our numerical experiments we have used:

- laptop MacBook Pro 13 (CPU: 2.7 GHz Intel Core i5; RAM: 16Gb)

- python's library scipy

### C.2.1 The case of $T = 2$

The maximum of the 3-variate functional $L_{D,\gamma^{\mathrm{s}},\gamma^{\mathrm{B}}}$ can be found in the hyperplane $v_2 = v_3$ (the proof is provided in Appendix C.1). Thus, for $T = 2$ the maximization problem is reduced[9] to a 2-variate optimization of the function $L_2 : \Delta^2 \to \mathbb{R}$, where $L_2(v_1, v_2) = (1 - F_D(\mathbf{v}))^{\mathsf{T}} \Upsilon_2(\gamma_{\mathsf{S}}, \gamma_{\mathsf{B}}) \mathbf{v}$, $\mathbf{v} \in \Delta^2$, and $\Upsilon_2(\gamma_{\mathsf{S}}, \gamma_{\mathsf{B}}) = \begin{pmatrix} \gamma_{\mathsf{S}} & 0 \\ -(\gamma_{\mathsf{S}} - \gamma_{\mathsf{B}}) & 1 + \gamma_{\mathsf{S}} - \gamma_{\mathsf{B}} \end{pmatrix}$.

Note that, for the uniform distribution $D = U[0; 1]$, $F_D(v) = v$ and the optimized functional $L_2$ becomes thus quadratic. Hence the problem can be solved by means of QP. We solve this problem numerically using the Sequential Least Squares Programming method. So, for several pairs of $(\gamma_{\mathsf{S}}, \gamma_{\mathsf{B}})$, we find the optimal algorithm $\mathcal{A}^*$ and depict in Fig. C.1 both its prices $\mathcal{A}^*(\mathfrak{n})$ for all nodes $\mathfrak{n}$ and its relative expected strategic revenue (w.r.t. $\mathcal{A}_1^*$). Namely, Fig. C.1(a) contains results for $\gamma_{\mathsf{S}} = 0.8$ and $\gamma_{\mathsf{B}} \in \{0.01 + i \cdot 0.005\}_{i=0}^{148}$, while Fig. C.1(b) contains results for $\gamma_{\mathsf{B}} = 0.2$ and $\gamma_{\mathsf{S}} \in \{0.2 + i \cdot 0.005\}_{i=0}^{159}$.

First, at the bottom of Fig. C.1 we see that *the optimal algorithm outperforms the baseline optimal constant pricing for any observed pair of discounts.* Second, the top part of Fig. C.1 demonstrates us that, for any pair of discounts, the optimal algorithm is a *consistent pricing*, i.e., the one which never sets prices lower (higher) than earlier accepted (rejected, resp.) ones [2]. In fact, this property is theoretically guaranteed for the studied case; namely, it easily follows from the relation between the optimal prices and the optimum $\mathbf{v}$: $\mathcal{A}^*(\mathbf{0}) = v_1$, $\mathcal{A}^*(\mathfrak{c}) = \gamma_{\mathsf{B}} v_1 + (1 - \gamma_{\mathsf{B}}) v_2$, and $\mathcal{A}^*(\mathbf{1}) = v_3$. Third, the obtained optimal algorithms are appeared to be continuous in $\gamma_{\mathsf{S}}$ and $\gamma_{\mathsf{B}}$. Moreover, if the distance between the discount rates $\gamma_{\mathsf{S}}$ and $\gamma_{\mathsf{B}}$ converges to 0, then the optimal algorithm $\mathcal{A}^*$ converges to the optimal constant one $\mathcal{A}_1^*$.

**Figure C.1:** 2-round game. The prices $\mathcal{A}^*(\mathbf{0}), \mathcal{A}^*(\mathfrak{c}), \mathcal{A}^*(\mathbf{1})$ and the relative expected strategic revenue (w.r.t. $\mathcal{A}_1^*$) of the optimal algorithm $\mathcal{A}^*$ for discount rates: (a) $\gamma_{\mathsf{S}} = 0.8$ and various $\gamma_{\mathsf{B}}$; (b) $\gamma_{\mathsf{B}} = 0.2$ and various $\gamma_{\mathsf{S}}$.

## C.2.2 The case of $T = 3$

In a similar way as it done for the previous case, the dimensionality of the optimization problem can be lowered from 7 to 4, when $\gamma_B < (\sqrt{5} - 1)/2$, and to 5, when $\gamma_B > (\sqrt{5} - 1)/2$[10]. The method to solve the optimization problem and the set of $(\gamma_S, \gamma_B)$ are the same as in the case of $T = 2$. Fig. C.2 is arranged similarly to Fig. C.1.

Analogously to the case of $T = 2$, in Fig. C.2, we observe the superiority of the optimal algorithm $\mathcal{A}^*$ over the baseline $\mathcal{A}_1^*$ for any pair of discount rates, as well as convergence to $\mathcal{A}_1^*$ as $|\gamma_S - \gamma_B| \to 0$ and the continuity of $\mathcal{A}^*$ in $\gamma_S$ and $\gamma_B$. But, in contrast to the case of $T = 2$, *the optimal algorithm may be non-consistent*: the condition of consistency is violated by the reverse order of the prices $\mathcal{A}^*(\mathfrak{e}) < \mathcal{A}^*(\mathtt{01})$ for $\gamma_B > \approx 0.54$ (which seen in Fig. C.2(a)), i.e., the seller offers a price larger than the one at the first round if the buyer rejects the first price, but accepts the one at the second round.

There is a lot of other interesting observations: e.g., pairs of equal prices when $\gamma_B \to 0$ (see Fig. C.1 and C.2); some specific area of pairs of $(\gamma_S, \gamma_B)$ where algorithm prices becomes equal (see Fig. C.2), etc. They are seen also in Sec.5, Fig.1 as well, and a thorough study of them is deferred to future work.

**Figure C.2:** 3-round game. The prices $\mathcal{A}^*(\mathfrak{n})$, for nodes $\mathfrak{n} \in \mathfrak{N}$ s.t. $|\mathfrak{n}| \leq 2$, and relative expected strategic revenue (w.r.t. $\mathcal{A}_1^*$) of the optimal algorithm $\mathcal{A}^*$ for discounts: (a) $\gamma_S = 0.8$ and various $\gamma_B$; (b) $\gamma_B = 0.2$ and various $\gamma_S$.

## C.2.3 The case of $T = \infty$: more details on observations

Several interesting facts, that reoccur for all distributions that we examined:

1. At the bottom of Fig. C.8, we see that *the optimal $\tau$-step algorithms $\mathcal{A}_\tau^*$ outperform the baseline optimal constant pricing $\mathcal{A}_1^*$ for any observed pair of discounts*. Moreover, Fig. C.8 demonstrates that the significant increase in revenue can be obtained even when the minimal possible step aside from the constant pricing is made ($\tau = 2$). For instance, the seller can extract up to $+20\%$ revenue by just maximizing the functional Eq. (4) in the 3-dimensional

space (since $2^\tau - 1 = 3$ for $\tau = 2$): e.g., the revenue improvement is larger than 20% for $\gamma_\mathsf{S} = 0.9, \gamma_\mathsf{B} = 0.2$, larger than 16% for $\gamma_\mathsf{S} = 0.8, \gamma_\mathsf{B} = 0.5$, and larger than 10% for $\gamma_\mathsf{S} = 0.8, \gamma_\mathsf{B} = 0.55$.

2. We see that the expected strategic revenue of $\mathcal{A}_\tau^*$ converges quite quickly to the optimal one (which thus larger than the revenue of the baseline $\mathcal{A}_1^*$ as well). This observation constitutes the empirical evidence of Prop. 3, which suggests that the convergence rate is equal to $\gamma_\mathsf{S}$.

3. The top part of Fig. C.8 demonstrates us that *an optimal algorithm may be non-consistent*[11]: e.g., the condition of consistency is violated by the reverse order of the prices $\mathcal{A}_4^*(\mathfrak{e}) < \mathcal{A}_4^*(\mathtt{001})$ for $\gamma_\mathsf{B} > \approx 0.57$ (which is seen in Fig. C.8(a)), i.e., the seller offers a price larger than the one at the first round if the buyer rejects the first and second prices, but accepts the one at the third round.

4. The obtained optimal algorithms are appeared to be continuous in $\gamma_\mathsf{S}$ and $\gamma_\mathsf{B}$. Moreover, if the distance between the discount rates $\gamma_\mathsf{S}$ and $\gamma_\mathsf{B}$ converges to 0, then the optimal algorithm $\mathcal{A}^*$ converges to the optimal constant one $\mathcal{A}_1^*$ (what experimentally supports that $H_D$ is a special case of $L_{D,\gamma^\mathsf{S},\gamma^\mathsf{B}}$).

5. There is a lot of other interesting observations (see Fig. C.8): pairs of equal prices when $\gamma_\mathsf{B} \to 0$; some specific area of pairs of $(\gamma_\mathsf{S}, \gamma_\mathsf{B})$ where algorithm prices becomes equal; etc. A thorough study of them is left for future work.

Overall, we conclude that *learning of prices even in several starting rounds allow to extract revenue significantly larger than the one of optimal static pricing.*

### C.2.4 Different distributions

Here we provide plots for $V \sim \beta(4, 2), V \sim \beta(2, 4)$, and $V$ distributed with the density $\frac{1 - e^{-x}}{1 - e^{-1}} \mathbb{I}_{0 \leq x \leq 1}$ in different special cases. Discounts are taken identically to those from Appendices C.2.1 and C.2.2 (as well as grids for $\gamma_\mathsf{B}$ and $\gamma_\mathsf{S}$). Figures descriptions are given in the following list:

1. Figure C.3: 2-round game for $V \sim \beta(4, 2)$.

2. Figure C.4: 2-round game for $V \sim \beta(2, 4)$.

3. Figure C.5: 3-round game for $V \sim \beta(4, 2)$.

4. Figure C.6: 3-round game for $V \sim \beta(2, 4)$.

5. Figure C.7: infinite game for $V$ distributed with the density $\frac{1 - e^{-x}}{1 - e^{-1}} \mathbb{I}_{0 \leq x \leq 1}$, which is a density of a random variable $\xi \sim Exp(1)$ conditioned by $0 \leq \xi \leq 1$.

6. Figure C.8: infinite game for $V \sim U[0, 1]$ (uniform distribution).

**Figure C.3:** 2-round game, $V \sim \beta(4, 2)$. The prices $\mathcal{A}^*(\mathtt{0}), \mathcal{A}^*(\mathfrak{e}), \mathcal{A}^*(\mathtt{1})$ and the relative expected strategic revenue (w.r.t. $\mathcal{A}_1^*$) of the optimal algorithm $\mathcal{A}^*$ for discount rates: (a) $\gamma_{\mathsf{S}} = 0.8$ and various $\gamma_{\mathsf{B}}$; (b) $\gamma_{\mathsf{B}} = 0.2$ and various $\gamma_{\mathsf{S}}$.

**Figure C.4:** 2-round game, $V \sim \beta(2, 4)$. The prices $\mathcal{A}^*(\mathtt{0}), \mathcal{A}^*(\mathfrak{e}), \mathcal{A}^*(\mathtt{1})$ and the relative expected strategic revenue (w.r.t. $\mathcal{A}_1^*$) of the optimal algorithm $\mathcal{A}^*$ for discount rates: (a) $\gamma_{\mathsf{S}} = 0.8$ and various $\gamma_{\mathsf{B}}$; (b) $\gamma_{\mathsf{B}} = 0.2$ and various $\gamma_{\mathsf{S}}$.

**Figure C.5:** 3-round game, $V \sim \beta(4, 2)$. The prices $\mathcal{A}^*(\mathfrak{n})$, for nodes $\mathfrak{n} \in \mathfrak{N}$ s.t. $|\mathfrak{n}| \leq 2$, and relative expected strategic revenue (w.r.t. $\mathcal{A}_1^*$) of the optimal algorithm $\mathcal{A}^*$ for discounts: (a) $\gamma_{\mathtt{S}} = 0.8$ and various $\gamma_{\mathtt{B}}$; (b) $\gamma_{\mathtt{B}} = 0.2$ and various $\gamma_{\mathtt{S}}$.

**Figure C.6:** 3-round game, $V \sim \beta(2, 4)$. The prices $\mathcal{A}^*(\mathfrak{n})$, for nodes $\mathfrak{n} \in \mathfrak{N}$ s.t. $|\mathfrak{n}| \leq 2$, and relative expected strategic revenue (w.r.t. $\mathcal{A}_1^*$) of the optimal algorithm $\mathcal{A}^*$ for discounts: (a) $\gamma_{\mathtt{S}} = 0.8$ and various $\gamma_{\mathtt{B}}$; (b) $\gamma_{\mathtt{B}} = 0.2$ and various $\gamma_{\mathtt{S}}$.

**Figure C.7:** Infinite game $T = \infty$, $V$ distributed with the density $\frac{1-e^{-x}}{1-e^{-1}}\mathbb{I}_{0\leq x\leq 1}$. The prices $\mathcal{A}_4^*(\mathfrak{n})$, for nodes $\mathfrak{n} \in \mathfrak{N}$ s.t. $|\mathfrak{n}| \leq 3$, of the optimal 4-step algorithm $\mathcal{A}_4^*$ and the relative expected strategic revenue (w.r.t. $\mathcal{A}_1^*$) of the optimal $\tau$-step algorithm $\mathcal{A}_\tau^*, \tau = 2, .., 6$, for discounts: (a) $\gamma_{\mathsf{S}} = 0.8$ and various $\gamma_{\mathsf{B}}$; (b) $\gamma_{\mathsf{B}} = 0.2$ and various $\gamma_{\mathsf{S}}$.

**Figure C.8:** Infinite game $T = \infty$, uniform $D$. The prices $\mathcal{A}_4^*(\mathfrak{n})$, for nodes $\mathfrak{n} \in \mathfrak{N}$ s.t. $|\mathfrak{n}| \leq 3$, of the optimal 4-step algorithm $\mathcal{A}_4^*$ and the relative expected strategic revenue (w.r.t. $\mathcal{A}_1^*$) of the optimal $\tau$-step algorithm $\mathcal{A}_\tau^*, \tau = 2, .., 6$, for discounts: (a) $\gamma_{\mathsf{S}}=0.8$ and various $\gamma_{\mathsf{B}}$; (b) $\gamma_{\mathsf{B}}=0.2$ and various $\gamma_{\mathsf{S}}$.

# D    Discussion on regularity

The regularity of the discount $\gamma^{\mathsf{B}}$ is used in two cases, namely, to get: (1) the uniqueness of $\gamma$-dependent natural order of the strategies $\mathfrak{S}$; (2) zero probability of the set of the valuations for which the optimal buyer strategy is not unique. The case (1) is used in Lemma A.1 and Prop. 2 from Sec.4; there, regularity is just needed for simplicity of presentation of the proofs; these statements (possibly with a slight change) will certainly hold without this restriction on $\gamma^{\mathsf{B}}$. The case (2) is used in Prop. 1 to guarantee that the strategic buyer will not prefer (with non-zero probability) a strategy that has been non-active before the transformation. So, Prop. 1 may not hold without regularity of $\gamma^{\mathsf{B}}$. But we believe that one can obtain a similar result for a series of algorithms that "converges" to a one from $\tilde{\mathfrak{A}}$ and use this series to obtain the statement of Th. 3. In any way, the restriction on the regularity of $\gamma^{\mathsf{B}}$ does not harm the main conclusions of our work, because, for a finite horizon, regular discounts are more frequent than non-regular ones, e.g., there is just a finite number of non-regular geometric discounts for a finite horizon. Hence, our qualitative results from Sec.5 and Appendix C are not affected by this restriction.

# E    Our setting as two-stage game

Due to the commitment and the presence of only one buyer, our setting of repeated posted-price auctions can be formalized as a two stage game.

The common knowledge here are the discounts $\gamma^{\mathsf{B}}$, $\gamma^{\mathsf{S}}$, and the prior distribution $D$ of the private valuation $V$, while the realization $v$ of $V$ is known only by the buyer. At the first stage, the seller picks a pricing algorithm $\mathcal{A} \in \mathfrak{A}_T$, her choice is announced to the buyer; at the second stage, the buyer picks a buyer strategy $\mathbf{a} \in \mathfrak{S}_T$. The buyer's utility is the surplus and the seller's one is the expected revenue (see Eq. (1)). Thus, if some pricing $\mathcal{A}^* \in \mathfrak{A}_T$ is a solution to our problem, then the pair $(\mathcal{A}^*, \mathbf{a}^{\mathrm{Opt}}(\mathcal{A}^*, v, \gamma^{\mathsf{B}}))$ will be an equilibrium of above described game.

We believe that an equilibrium concept is unnatural for our setting, and our approach to formalize the problem is more suitable.

Also note that the backward induction isn't applicable here. There is no way to find an optimal left subtree without fixing the root and the right subtree and vice versa, since the set of types that rejects (accepts) the root price depends on all of the three. Note that backward induction is useful when there are long enough sequences of subgames embedded one into another in the game, while in our problem the game is essentially two-stage.

# F    Example of a real instantiation of the studied auction setup in web advertising industry

During our work on the paper, we considered real practical problems of one of the most popular global ad exchange. This Internet company faces with instances of our game, that can be described by the following example: an Internet user searches for an apartment for rent; an advertiser (with an ad about apartments) targets this user. An ad exchange (seller) tracks this user each time she visits web sites related to the rent intent. Each view of a web page with a vacant ad slot by this user is a round (t=1,2,..) in a sequence of posted price auctions between the seller and the advertiser (buyer). The advertiser holds fixed valuation $v$ for a view of this user of his ad about apartments until the user holds the rent intent. The discount rate $\gamma$ is the probability that the user will still search for an apartment for rent at the next round.

In practice of ad exchanges, many thousands of instances of our game are performed each day. In this case, the buyer believes that the seller will follow the committed algorithm, since the seller does it all previous instances. On the other hand, the seller is incentivized to follow her commitment, since its violation will incur significant losses due to loss of trust from advertisers, because the studies [9, 1, 6] showed that the seller earns noticeably less revenue in settings without commitment than with it. In particular, when the seller is faced with one buyer and does not commit for a pricing algorithm, the buyer in perfect Bayesian equilibrium (PBE) rejects goods all rounds except a low number of last ones. These studies constitute economic arguments for the seller to commit for a pricing algorithm and to do the best to assure the buyer that the commitment will not be violated in practice.

# G    Lower bound on revenue improvement for the case $\gamma_S = 1 > \gamma_B$

In the special case of the more patient seller with $\gamma_S = 1$, one can get theoretically guaranteed lower bound on the optimal expected strategic revenue and thus mathematically prove that the constant Myerson pricing is no longer optimal.

**Lemma G.1.** *In our setup, the following inequality holds:*

$$\mathbb{E}_{V \sim D}[V] > \max_{p \in \mathbb{R}_+} H_D(p). \tag{G.1}$$

*Proof.* By the definition we have:

$$\mathbb{E}_{V \sim D}[V] = \int_0^{+\infty} v \, dF_D(v) \geq \int_p^{+\infty} v \, dF_D(v) > \int_p^{+\infty} p \, dF_D(v) = p(1 - F_D(p)) \quad \forall p \in \mathbb{R}_+.$$

Since, by the definition, $H_D(p) = p(1 - F_D(p))$, we obtain the claimed inequality.  □

**Proposition G.1.** *Let the valuation $V$ be distributed on $[0,1]$ (i.e., $F_D(1) = 1$), the discounts be $\gamma_t^B = \gamma_B^{t-1} \mathbb{I}_{\{t \leq T\}}$, $\gamma_B \in (0,1)$, and $\gamma_t^S = \mathbb{I}_{\{t \leq T\}}$ (i.e., a game with a finite horizon $T \in \mathbb{N}$ is considered and the seller's strategic revenue is without discount). Then, if $\mathcal{A}_1^*$ is the optimal constant algorithm (i.e., offers always the Myerson price $p_D^*$) and $\mathcal{A}^*$ be the optimal algorithm, then the following lower bound holds for the optimal expected strategic revenue:*

$$\mathbb{E}_{V \sim D}[\mathrm{SRev}_{\gamma^S, \gamma^B}(\mathcal{A}^*, V)] \geq T \mathbb{E}_{V \sim D}[V] - (r \mathbb{E}_{V \sim D}[V] + 4)(\log_2 \log_2 T + 2), \tag{G.2}$$

*where $r := \lceil \log_{\gamma_B}\left((1 - \gamma_B)/2\right) \rceil$, and the following bound for relative gain in the expected revenue of the optimal algorithm $\mathcal{A}^*$ w.r.t. the optimal constant pricing $\mathcal{A}_1^*$ holds as well:*

$$\frac{\rho_{\mathcal{A}^*}}{\rho_{\mathcal{A}_1^*}} \geq \frac{\mathbb{E}_{V \sim D}[V]}{H_D(p_D^*)} - \frac{r \mathbb{E}_{V \sim D}[V] + 4}{H_D(p_D^*)} \cdot \frac{\log_2 \log_2 T + 2}{T} \xrightarrow[T \to +\infty]{} \frac{\mathbb{E}_{V \sim D}[V]}{H_D(p_D^*)} > 1. \tag{G.3}$$

*where the following short notation is used: $\rho_{\mathcal{A}} := \mathbb{E}_{V \sim D}[\mathrm{SRev}_{\gamma^S, \gamma^B}(\mathcal{A}, V)]$.*

*Proof.* Let us consider the Penalized Reject-Revising Fast Exploiting Search [2] (PRRFES) as a pricing algorithm, for which the following bound holds [2, Th.5]:

$$Tv - \mathrm{SRev}_{\gamma^S, \gamma^B}(\mathcal{A}_{\mathrm{PRRFES}}, v) \leq (rv + 4)(\log_2 \log_2 T + 2) \qquad \forall v \in [0,1] \quad \forall T \geq 4. \tag{G.4}$$

Eq. (G.4) implies Eq. (G.2), because

$$\mathbb{E}_{V \sim D}[\mathrm{SRev}_{\gamma^S, \gamma^B}(\mathcal{A}^*, V)] \geq \mathbb{E}_{V \sim D}[\mathrm{SRev}_{\gamma^S, \gamma^B}(\mathrm{PRRFES}, V)] \geq$$
$$\geq T \mathbb{E}_{V \sim D}[V] - (r \mathbb{E}_{V \sim D}[V] + 4)(\log_2 \log_2 T + 2). \tag{G.5}$$

In order to obtain the first inequality in Eq. (G.3), one needs Eq. (G.2) and remind that $\mathbb{E}_{V \sim D}[\mathrm{SRev}_{\gamma^{\mathrm{S}}, \gamma^{\mathrm{B}}}(\mathcal{A}_1^*, V)] = TH_D(p_D^*)$.

Finally, the inequality $\frac{\mathbb{E}_{V \sim D}[V]}{H_D(p_D^*)} > 1$ holds due to Lemma G.1.

$\square$

So, Eq. (G.2) is a theoretical guarantee of the fact that the constant optimal pricing is no more optimal in the considered case, because the relative improvement lower bound converges to the constant which is strictly higher than 1. Therefore, this result supports our empirical evidence from the numerical experiments. Moreover the provided improvement is achievable by the algorithm PRRFES [2].

In order to demonstrate the possible relative improvement, let us consider the uniform distribution on $[0, 1]$ as an example of $D$, for which the Myerson price $p_D^* = 1/2$ and the expected strategic revenue provided by the optimal constant algorithm $\mathcal{A}_1^*$ is in turn $\mathbb{E}[\mathrm{SRev}_{\gamma^{\mathrm{S}}, \gamma^{\mathrm{B}}}(\mathcal{A}_1^*, V)] = T/4$.

Based on Proposition G.1, we have

$$\frac{\rho_{\mathrm{PRRFES}}}{\rho_{\mathcal{A}_1^*}} \geq 2 - (2r + 16)\frac{\log_2 \log_2 T + 2}{T}, \tag{G.6}$$

which goes to 2 as $T \to \infty$. Hence, we see that the seller is able to increase his revenue by up to $+100\%$ (depending on the horizon $T$) just by applying the algorithm PRRFES instead of the optimal constant one. Note that PRRFES is horizon independent, and the seller thus may not know the horizon $T$ in advance to apply it.

# H    Auxiliary definitions and propositions

## H.1    Absolute continuity of the Lebesgue integral ([7])

**Proposition H.1.** *Consider the Lebesgue measure $\mu$ on $\mathbb{R}$ and let $f : \mathbb{R} \to \mathbb{R}$ be an integrable function on $A \subset \mathbb{R}$, then for any $\varepsilon > 0$ there exists such $\delta > 0$ that*

$$\left| \int_B f(x)d\mu \right| < \varepsilon,$$

*where $B \subset A : \mu(B) < \delta$.*

# I Pseudo-code for evaluating the matrix $\Xi_T(\boldsymbol{\gamma}^{\mathsf{S}}, \boldsymbol{\gamma}^{\mathsf{B}})$.

---

**Algorithm I.1** Evaluation of a pair of matrices $(K_T(\gamma, \gamma), K_T(\gamma, \gamma'))$

---

1: **Input:** $\gamma, \gamma' \in \mathbb{R}_+$ and $T \in \mathbb{N}$
2: **if** $T = 1$ **then**
3:   $(K, K') := ((1), (1))$
4: **else**
5:   $(K_{T-1}, K'_{T-1}) := \mathrm{ALG}I.1(\gamma, \gamma', T-1)$
6:   $k := 2^T - 1, k_{-1} := 2^{T-1} - 1$
7:   $(\hat{K}, \hat{K}') := \left( \begin{pmatrix} \gamma K_{T-1} & 0_{k_{-1} \times 1} & 0_{k_{-1} \times k_{-1}} \\ 0_{1 \times k_{-1}} & 1 & 0_{1 \times k_{-1}} \\ 0_{k_{-1} \times k_{-1}} & 1_{k_{-1} \times 1} & \gamma K_{T-1} \end{pmatrix}, \begin{pmatrix} \gamma' K'_{T-1} & 0_{k_{-1} \times 1} & 0_{k_{-1} \times k_{-1}} \\ 0_{1 \times k_{-1}} & 1 & 0_{1 \times k_{-1}} \\ 0_{k_{-1} \times k_{-1}} & 1_{k_{-1} \times 1} & \gamma' K'_{T-1} \end{pmatrix} \right)$
8:   **for** $i = 1, \dots, 2^T - 1$ **do**
9:     $w_i := \sum_{j=1}^{2^T - 1} \hat{\kappa}_{ij}$, where $\hat{K} = (\hat{\kappa}_{ij})$
10:   **end for**
11:   Find the permutation $\pi$ s.t. $w_{\pi(1)} < w_{\pi(2)} < \cdots < w_{\pi(2^T-1)}$.
12:   $(K, K') := (\pi(\hat{K}), \pi(\hat{K}'))$, where $\pi(A)$ is the matrix $A$ with rows permuted according to $\pi$.
13: **end if**
14: **Output:** $(K, K')$

---

---

**Algorithm I.2** Evaluation of the matrix $\Xi_T(\gamma_{\mathsf{B}}, \gamma_{\mathsf{S}})$

---

1: **Input:** $\gamma, \gamma' \in \mathbb{R}_+$ and $T \in \mathbb{N}$
2: $J := (\eta_{ij})_{i,j \in [2^T - 1]}$, where

$$\eta_{ij} = \begin{cases} 1, & \text{if } i = j \\ -1, & \text{if } i = j + 1 \\ 0, & \text{otherwise} \end{cases}$$

3: $I := (\iota_{ij})_{i,j \in [2^T - 1]}$, where

$$\iota_{ij} = \begin{cases} 1, & \text{if } i \geq j \\ 0, & \text{otherwise} \end{cases}$$

4: $(K, K') := \mathrm{ALG}I.1(\gamma_{\mathsf{B}}, \gamma_{\mathsf{S}}, T)$
5: **for** $i = 1, \dots, 2^T - 1$ **do**
6:   $w_i := \sum_{j=1}^{2^T - 1} \kappa_{ij}, w_0 := 0$, where $K = (\kappa_{ij})$.
7: **end for**
8: $M := (\mu_{ij})_{i,j \in [2^T - 1]}$, where

$$\mu_{ij} = \begin{cases} w_i - w_{i-1}, & \text{if } i = j \\ 0, & \text{otherwise} \end{cases}$$

9: $\Xi := J \cdot K \cdot (K')^{-1} \cdot I \cdot M$
10: **Output:** $\Xi$

---

## Footnotes

[1]Details on feasible mechanisms in [8, Sec.3].

[2]This price can be find by the equation $p = (1 - F_D(p))/f_D(p)$, when $D$ has continuous probability density $f_D$.

[3]E.g., a consistent order: the nodes from the left subtree come before the root node $\mathfrak{e}$, and the ones from the right subtree come after the root $\mathfrak{e}$; then we recursively repeat this rule for the left and right subtrees.

[4] In other words, the node $\mathfrak{n}_j$ can be represented in the string notation as $a^i_1 \ldots a^i_{t-1}$ for some $1 \leq t \leq T$.

[5]This fact is trivial for matrices $Z_T$ and $J_T$. To show this for $K_T$, just apply the induction. By rearranging of rows and columns of $K_T$ (it does not affect the property of invertibility) one can obtain a block diagonal matrix with two blocks. Each of these blocks is based on a matrix with the form like $K_{T-1}$.

[6] Here we allow discounts to be zero, but this does not affect the result.

[7]Remember Def.2 from Sec.4: $\mathbf{a} \cdot \mathbf{b}$ is for the scalar product.

[9]This case shows that even though the dimension of the problem can be reduced, it still could not be reduced to a one-dimensional problem in general. The same we observe in the case of $T = 3$.

[10]The different cases are results of the change of the order of the values $\{\boldsymbol{\gamma}^B \cdot \mathbf{a} | \mathbf{a} \in \mathfrak{S}\}$ at the border point $(\sqrt{5} - 1)/2$.

[11]A consistent algorithm never sets prices lower (higher) than earlier accepted (rejected, resp.) ones [2, 3, 4, 5].