[Reviews · NeurIPS 2019]

Reviewer 1



This work studies optimal pricing is repeated posted-price auctions. They consider the scenario in which a single seller interacts with a single buyer whose patience is different than that of the seller. The main contribution of this work is designing optimal pricing algorithms to maximize the revenue of the seller. They consider two different cases which are selling to a buyer who is more patient than the seller and selling to a buyer who is less patient. They show that the former case is simpler and they provide an algorithm called the "big deal" algorithm for that. To get this result, they first show how in the case of equal discounts (patience) Myerson's well-known result can be used to obtain the optimal revenue. Then, they use it to solve the case of a less patient seller. Further, for the case of a less patient buyer, they give a more complicated algorithm and to obtain that, they use a reduction to the optimization of a multivariant function. Strengths: --- 1- An interesting problem that is not studied before 2- A well-written paper 3- Paper has a good technical contribution and authors verify some of their claims using practical evidence. Weaknesses: --- My main problem is with the assumption that the seller has an accurate estimate of the buyer's discount and with the fact that it is not explicitly mentioned in the problem statement. This specifically concerns me because the "Big deal" pricing algorithm seems to highly rely on this assumption. As I understand if the seller underestimates buyer's patience and uses this algorithm, it might lead to a zero revenue for him.

Reviewer 2



1. In the preliminaries, it is not clear whether the discount sequences of the buyer and the seller are public information or not. In the later section, in my understanding, the discount sequences are known for both buyer and seller. But I think this assumption is slightly unrealistic, since this means the seller knows the extent of impatience of the buyer. A more reasonable assumption might be that the seller knows the distribution of the discount parameter of the buyer. 2. In line 100, “A*’s ESR is maximal”, is it “maximum”? 3. In section 3, when the seller is less patient, the result in the paper indicates the repeated game is meaningless since the best algorithm is a “once-for-all” deal. It does not seem to match the real case. The reason might be that the seller does not know the exact discount of buyer, which is related to the first comments. 4. In section 4, the results require that the discount is regular. Should the discount also decrease geometrically? 5. In section 4, about the CA algorithm, I understand that any algorithm can be transformed to be a CA algorithm, but I am not quite sure about the intuition behind the CA algorithm. For a CA algorithm, does it mean (in some sense) that there is no “stupid” buyer strategy where any strategy is optimal for at least one valuation? 6. In line 231, “which is piecewise linear”, according to the following argument, is it “piecewise constant”? 7. In section 5, the approximation algorithm looks weak and straightforward. The paper shows that it is a good approximation to consider only finite steps. But the algorithm for t steps needs poly(2^t) times which seems unrealistic in the practice. Is it possible to show some hardness result for the case of t-step pricing algorithm? Such result might be a good justification of the approximation algorithm After the feedback, 1. In the feedback, the authors answer the question about the discount knowledge and provide further explanations about the unknown buyer discount. I think the explanation is good and should be add into the full version of paper. 2. But the feedback about the time complexity is unsatisfying. Since the space complexity is exponential on \tau, the time complexity is also exponential which means only very small constant \tau is tolerable. Then the authors should justify why it is reasonable in the practice.

Reviewer 3



The paper studies the repeated posted-price setting against a single strategic buyer with fixed valuation and a known prior over that valuation, and asymmetric discounting. They give the optimal algorithm when the seller is less patient than the buyer, which is simple and straight-forward but insightful. When the seller is more patient than the buyer, they provide an interesting characterization seller algorithms as a piece-wise linear surplus function over buyer valuations. The final results here are less clean. The authors avoid giving an explicit characterization of the optimal algorithm, but instead suggest an (exponential) procedure for computing the optimal seller algorithm. “The case of our setting where the time discounts of the seller and 42 the buyer are different was never studied before.” This is studied in the referenced works [7], and [8], specifically where the buyer is less patient than the seller. Perhaps the authors mean that this has previously only been studied in the worst-case setting. Line 75: It is a little strange to call a sequence a strategy. A strategy is typically a function of the information received on previous rounds, although maybe this distinction is unimportant since A is deterministic? Line 77: “Given an algorithm A … is uniquely determined.” If this is the case, seller algorithm is constrained to be deterministic, which could be restrictive. This should be stated earlier. Line 99: “Expected” does not make sense as there is no expectation in (1). Line 125: “How greater the maximal ESR” => typo Line 158: typo: Myeson Line 171: The algorithm requires the seller to know gamma_B. Line 211: “tangent” here is a little unclear. Do you mean equal at the valuation v? Lines 213-227: There seems to be something wrong with this construction. Consider, for example, the big deal algorithm from the beginning of the paper. This algorithm is not completely active. Let a be a buyer strategy that accepts the initial big deal, but then rejects some of the free items that come afterwards. There is no valuation that makes this surplus-optimal. However, A(n) cannot be decreased any further (it is already zero). Am I missing something?

Reviewer 4



The authors consider the problem of coming up with a pricing strategy that is god against a strategic buyer. Here, a strategic buyer with a fixed valuation v is one who knows the pricing strategy used by a seller and acts in the auction to optimize her surplus. The authors consider an scenario where both the buyer and the seller have a discount factor in their surplus and revenue. Similar scenarios have been studied before by Kamin et al., Mohri and Munoz and Dustra et al. However, these works have focused on the regret achievable when facing a single buyer. In contrast, this paper wants to find an optimal pricing strategy that works well on average assuming the value of a buyer is chosen from a fixed (known) distribution. The paper describes the set of optimal strategies when the discount factor of the buyer is less than that of the seller and vice-versa. The first case offers surprising results showing that the buyer can recover the same revenue as if he had a discount factor equal to that of the buyer. The case when the seller's discount greater than the buyer discount is much more complicated as it requires optimizing over a discrete set with an exponential number of elements. I would say the main contribution of the authors is to transform this problem into a smooth problem where gradients are available. This was by no means a trivial task and required a very deep understanding of the problem. While the smooth problem size remains exponential, the authors show that a low dimensional approximation to it does not deteriorate the solution too much. The authors conclude showing different performance metrics for their algorithm. Overall I am very impressed with the work done by the authors to obtain a smooth version of this problem and make a connection between strategic revenue optimization and static revenue optimization which was missing in the literature. One question that the authors do not address however is how does their approximation compare to using the algorithm of Drutsa et. al. While the algorithm of Drutsa et al. is designed for facing a fixed buyer, one can plug the guarantees inside the expectation term and see if they are better than the ones proposed by the authors. This would have the advantage of being a much simpler algorithm. Even if this were the case, however, I believe making the connection between strategic revenue optimization and static revenue optimization is a good enough reason to accept this paper. **** I must admit that I did not read the proofs of the paper although intuitively everything seemed correct ******

[Author Response · NeurIPS 2019]

We thank the reviewers and will do our best to improve the presentation. **To Reviewers #2 & #3 on discount knowledge**. Yes, in our setup, the buyer's $\gamma_B$ is public knowledge; we will clarify this in the text. Such scenario can arise in one of our model interpretations (Lines 114-119 & Appendix F), where an RTB platform (seller S) has more data than an advertiser (buyer B) and may know which data are not available to B. In this interpretation, $\gamma_B, \gamma_S$ are B's and S's estimates of a true discount factor $\gamma$, which is a random variable unknown to both B, S. Consider a toy example: $\gamma = \xi_1 + \xi_2$, S observes $\xi_1$ and $\xi_2$, while B observes only $\xi_1$.

ESR relative to optimal in case of estimation error; g_S = 0.5; v ~ U[0; 1]; T = 5;

Say, B's estimate for $\gamma$ is $\gamma_B = \xi_1 + \mathbb{E}[\xi_2]$ (we took the simplest estimate for illustration), then the seller S can evaluate $\gamma_B$.

**What if the seller doesn't know buyer's $\gamma_B$ exactly.** In fact, our results can be useful in such a scenario as well. Case (1): if the seller knows only a lower bound $\hat{\gamma}_B$ for $\gamma_B$ s.t. $\gamma_S < \hat{\gamma}_B$, then she can apply "Big deal", which prices are calculated using $\hat{\gamma}_B$: $\mathcal{A}_{bd}(e) = \sum_t \hat{\gamma}^B p_D^*$; $\mathcal{A}_{bd}(1 \circ \mathfrak{n}) = 0 \ \forall \mathfrak{n}$; $\mathcal{A}_{bd}(0 \circ \mathfrak{n}) = T p_D^* \ \forall \mathfrak{n}$. Buyer (whose discount $\gamma_B \geq \hat{\gamma}_B$) with valuation $v > p_D^*$ still accepts the first proposed price, hence, the seller gets at least $\sum_t \hat{\gamma}^B p_D^* (1 - F(p_D^*))$. This is less than the optimal revenue (when $\gamma_B$ is known exactly), but strictly larger than the one of static pricing. Similarly, modifications of "Big deal" can be applied when seller knows only distribution of $\gamma_B, \gamma_B \geq \gamma_S$.

Case (2): Seller uses functional $L$ to find an optimal algorithm, assumes buyer's discount is $\gamma_B' = \gamma_B + \varepsilon$, but faces a buyer with true discount $\gamma_B$. We evaluate the loss in revenue by the following numerical experimentation: $T = 5$, $V \sim U[0; 1]$ (uniform on $[0; 1]$) and $\gamma_S = 0.5$ (different sets of parameters give qualitatively the same results). In figure above, the expected strategic revenue (ESR) of this seller is divided by the ESR of a well-informed seller (i.e. s.t. $\varepsilon = 0$). We see: (a) if $\varepsilon$ is small enough (for $\varepsilon = 0.02$, or $\geq 4\%$ of $\gamma_B$), then S still able to extract over 99% of the optimal ESR; (b) even if $\varepsilon$ is very large (for $\varepsilon = 0.1$, or $\geq 20\%$ of $\gamma_B$) S still able to extract over 97% of the optimal ESR for most cases ($\gamma_B \leq 0.4$); and (c) if S is able to just separate $\gamma_B$ of $\gamma_S$ with a decent margin, then she is able to gain extra revenue.

To Reviewer #2.

**On time complexity.** Application of pricing algorithms has no time complexity issues, since a seller just needs to track the current node in a tree to post a price in a round, however she needs to have enough memory (for $2^T - 1$ float variables). As of numerical methods to optimize our functional $L(\cdot)$, it's took few seconds to converge in all our experiments.

To Reviewer #3.

**On point 4.** No. Geometric discounts are considered for sake of exposition, but our results hold for non-geometric discounts as well. They are studied in Appendices A.1 & A.2, as indicated in Remarks 1 & 2 (see Lines 189, 283-284).

**On point 6.** No. It is written correctly: $S(\cdot)$ is piecewise *linear*, because, in a piece (an interval $(v_i, v_{i+1})$) $S(\cdot)$ equals to $S_{\mathbf{a}^i}(\cdot)$ for some strategy $\mathbf{a}^i$ which is a linear function of $v$: $S_{\mathbf{a}^i}(v) = (\sum_t \gamma_t^B a_t^i)v - (\sum_t \gamma_t^B a_t^i p_t)$ (see Lines 211 & 81).

**On point 7.** We have no hardness result for the optimization problem in Eq. (5), but, generally, it does not have a closed form solution (as we argue in Lines 271-273). Numerical methods help in this case, but they are usually very sensitive to the dimensionality of the problem. This is why we address the problem of dimensionality reduction and show that our functional $L(\cdot)$ is also useful to find optimal pricing algorithms in low-dimensional spaces. Note that low-dim spaces can be obtained by different constraints (see Lines 323-335), not only by $\tau$-step algorithms (as in Lines 290-322).

**On point 2&5.** Yes, you are right here. We will fix and make clear the text in these lines.

To Reviewer #4. **On Lines 125, 158 & 211.** Yes, you are right here, we will improve text in these places.

**On Lines 213-227 (construction of Prop.1's proof).** The construction in these lines (and, thus, the proof of Prop.1) works for the case $\gamma_S \geq \gamma_B$ as indicated in the statement of Prop.1 (Line 228) and considered in Sec.4 as a whole. When you take an algorithm Big Deal (note that you port it from the opposite case of discounts) and consider in the case $\gamma_S \geq \gamma_B$, the procedure from Lines 213-227 can be applied: the only strategy that can be activated in the first step is $10^{T-1}$ (see Appendix A.2.1, 3rd paragraph: we activate a strategy that has lowest last 1 position). This activation is provided via decreasing the price $\mathcal{A}(e)$ and increasing all prices $\mathcal{A}(1), \mathcal{A}(10), \ldots, \mathcal{A}(10^{T-2})$. After this step more possibilities for further "activations" arise.

**On an explicit description of the algorithm.** Shortly, it is as follows. Remind: for static pricing, the optimal (Myerson) price can be found from maximization of $H_D(p) = p(1 - F_D(p))$. In our dynamic case, the optimal algorithm can be found similarly: (a) construct the matrix $\Xi$ (a code to calculate its elements is in Appendix I); (b) construct the functional $L(\cdot)$ from Eq. (4); (c) find a vector $\mathbf{v}^{Opt}$ s.t. it maximizes $L(\mathbf{v})$, e.g., numerically using derivatives of $L(\cdot)$ provided in Lines 277-278; (d) convert the vector $\mathbf{v}^{Opt}$ to the prices of the optimal algorithm by means of linear transformation $\mathbf{w}_{\gamma^B}^{-1}(\cdot)$, which is mentioned in Lines 255 & 249 and whose matrix is $K_T(\boldsymbol{\gamma}^B, \boldsymbol{\gamma}^B)^{-1} J_T^{-1} Z_T(\boldsymbol{\gamma}^B)^{-1}$ (see Appendix A.2.2).

**On Line 42.** Yes, we will clarify: different discounts have only been studied in other setting (worst-case one [7,8,30,31]).

**On Lines 75 & 77.** Yes, we study deterministic algorithms, and, hence, a decision sequence is called a strategy (as in [7,30,31,59]). Note: it is easy to show that algorithms proposed in Sec.3 are optimal among probabilistic ones as well.

**On Line 99.** The word "expected" does not correspond to Eq.(1), but it is a part of the notion "expected strategic revenue" which is the reference for the expression $\mathbb{E}_{V \sim D}[\text{SRev}_{\boldsymbol{\gamma}^S, \boldsymbol{\gamma}^B}(\mathcal{A}^*, V)]$ situated at the beginning of Line 100.

**On Line 171.** Yes, $\gamma_B$ is publicly known. See 1st answer for Rev.#2 & #3 about the case when $\gamma_B$ isn't known precisely.

[Meta-Review · NeurIPS 2019]

The paper analyzes the setting of a repeated posted-price auction, where the buyer and seller have differing discount factors (\gamma_S, \gamma_B) and explicitly develops algorithms and analysis for the two ordering of these parameters. The setting in interesting and of significant importance to real-world applications. Overall, the reviewers agree that the paper provides several results that make important progress in the setting. There are some aspects that should be updated before considering final: - The algorithm requires knowing the discount rate of the buyer, this needs to be discussed more carefully as done in the reviewer feedback. - The setting requires access to the value distribution D. As pointed out in the paper, a large volume of historical bidding data is available, but this is a record of *strategic* behavior and it is not immediately clear to me that a truthful value distribution can always be extracted. Clarification on this point would be great.